# THE CONVERGENCE OF VARIANCE EXPLODING DIFFUSION MODELS UNDER THE MANIFOLD HYPOTHESIS

## ABSTRACT

Variance Exploding (VE) based diffusion models, an important class of diffusion models, have empirically shown state-of-the-art performance in many tasks. However, there are only a few theoretical works on the VE-based models, and those works suffer from a worse forward process convergence rate $1/\text{poly}(T)$ than the $\exp(-T)$ results of Variance Preserving (VP) based models, where $T$ is the forward diffusion time. The slow rate is due to the Brownian Motion without the drift term and introduces hardness in balancing the different error sources. In this work, we design a new forward VESDE process with a small drift term, which converts data into pure Gaussian noise while the variance explodes. Furthermore, unlike the previous theoretical works, we allow the diffusion coefficient to be unbounded instead of a constant, which is closer to the SOTA VE-based models. With an aggressive diffusion coefficient, the new forward process allows a faster $\exp(-T)$ rate. By exploiting this new process, we prove the first polynomial sample complexity for VE-based models with reverse SDE under the realistic manifold hypothesis. Then, we focus on a more general setting considering reverse SDE and probability flow ODE simultaneously and propose the unified tangent-based analysis framework for VE-based models. In this framework, we prove the first quantitative convergence guarantee for SOTA VE-based models with probability flow ODE. We also show the power of the new forward process in balancing different error sources on the synthetic experiments to support our theoretical results.

## 1 INTRODUCTION

In recent years, diffusion modeling has become an important paradigm for generative modeling and has shown SOTA performance in image generation, audio synthesis, and video generation (Rombach et al., 2022; Saharia et al., 2022; Popov et al., 2021; Ho et al., 2022). The core of diffusion models are two stochastic differential equations (SDE): the forward and reverse processes. The forward process can be described by an intermediate marginal distribution $\{q_t\}_{t \in [0,T]}$, which converts the data $q_0$ to Gaussian noise $q_T$. There are two types of forward SDE (Song et al., 2020b): (1) Variance Preserving (VP) SDE and (2) Variance Exploding (VE) SDE. The VPSDE corresponds to an Ornstein-Uhlenbeck (OU) process, and the stationary distribution is $\mathcal{N}(0, \mathbf{I})$. The VESDE corresponds to the Brownian motion with a (deterministic) time change, which has an exploding variance. In earlier times, VP-based models (Ho et al., 2020; Lu et al., 2022) provide an important boost for developing diffusion models. Recently, Karras et al. (2022) unify VPSDE and VESDE formula and show that the optimal parameters correspond to VESDE. Furthermore, VE-based models achieve SOTA performance in multi-step (Kim et al., 2022; Teng et al., 2023) and one-step image generation (Song et al., 2023).

After determining the forward SDE, the models reverse the forward SDE and generate samples by running the corresponding reverse SDE (Ho et al., 2020; Lu et al., 2022) or probability flow ODE (PFODE) (Song et al., 2020a; Karras et al., 2022). Before running the reverse process, three things need to be chosen: (1) a tractable reverse beginning distribution; (2) discretization scheme; (3) estimation of the score function $\nabla \log q_t$ (Ho et al., 2020). These procedures are error sources of the convergence guarantee, and we need to balance these terms to obtain the quantitative guarantee.

Despite the empirical success of diffusion models, only a few works focus on the convergence guarantee of these models. Furthermore, many previous works (Chen et al., 2023d;a;c; Lee et al., 2023; Benton et al., 2023) focus on VPSDE, and the analysis for VESDE is lacking. When considering the

reverse beginning error, Lee et al. (2022) analyze a special VESDE with constant diffusion coefficient $\beta_t$ and reverse SDE and obtain a $1/\sqrt{T}$ guarantee. This guarantee is worse than VP-based models since the reverse beginning error is $\exp(-T)$ for VP-based models. The above reverse beginning error introduces hardness to balance three error sources to achieve a great sample complexity, as shown in Section 5. To deal with this problem, De Bortoli et al. (2021) introduce a coefficient $\alpha$ to balance the drift and diffusion term. When choosing $\alpha = 1/T$, $q_T$ is $\mathcal{N}(0, T\mathbf{I})$, and the reverse beginning error is $\exp\left(-\sqrt{T}\right)$. However, their framework only allows constant $\beta_t$, far from the SOTA choice (Karras et al., 2022), whose coefficient grows fast and is unbounded. Furthermore, De Bortoli et al. (2021) fail to balance the above three errors and has exponential dependence on problem parameters. Therefore, the following question remains open:

*Is it possible to design a VESDE with a faster forward convergence rate than $1/poly(T)$ and balance error sources to achieve the polynomial complexity when the diffusion coefficient is unbounded?*

In this work, for the first time, we propose a new forward VESDE, which allows unbounded coefficients and enjoys a faster forward convergence rate. We first show that the new process with a conservative $\beta_t$ has similar trends but performs better compared to the original VESDE on synthetic data (Section 7). Then, we prove that the new process with a suitable aggressive $\beta_t$ can balance reverse beginning, discretization and score function errors and achieve the first polynomial sample complexity for VE-based models. It is worth emphasizing that our results are achieved under the manifold hypothesis, which means the data $q_0$ is supported on a lower dimensional compact set $\mathcal{M}$. Different from the previously discussed works, which assume the score function is Lipschitz continuous (Chen et al., 2023e) or satisfy strong log-Sobelev inequality (LSI) assumption (Lee et al., 2022; Wibisono and Yang, 2022) to compulsorily guarantee that the score does not explode when $t \to 0$, the manifold hypothesis allows the expansion phenomenon of the score function (Kim et al., 2021; Pope et al., 2021). To our knowledge, only Pidstrigach (2022) analyze VESDE in the continuous reverse process under the manifold hypothesis, which misses the key discretization step.

Besides the polynomial sample complexity for VE-based models, another important point for VE-based models is the quantitative guarantee for VESDE with reverse PFODE. Recently, many works show that models with reverse PFODE have faster sample rate (Lu et al., 2022), and VE-based models with reverse PFODE (Song et al., 2020b; Karras et al., 2022) achieve great performance. Furthermore, the reverse PFODE can be applied in areas such as computing likelihoods (Song et al., 2020b) and one-step generation models (Salimans and Ho, 2022; Song et al., 2023). However, except for a few theoretical works for VPSDE (Chen et al., 2023e;c), the current works focus on reverse SDE. Hence, after achieving polynomial complexity for reverse SDE, we go one step further and propose the tangent-based unified framework for VE-based models, which contains reverse SDE and PFODE. We note that when considering VPSDE with reverse SDE, Bortoli (2022) also use the tangent-based method. However, as discussed in Section 6.2, the original tangent-based lemma can not deal with reverse PFODE even in VPSDE. We carefully control the tangent process to avoid additional $\exp(T)$ by using the exploding variance property of VESDE. Using this unified framework, we achieve the first quantitative convergence for the SOTA VE-based models with reverse PFODE (Karras et al., 2022). In conclusion, we accomplish the following results under the manifold hypothesis:

1. We propose a new forward VESDE with the unbounded coefficient $\beta_t$ and a drift term that is typically small. With an aggressive $\beta_t$, the new process balances different error terms and achieves the first polynomial sample complexity for VE-based methods with reverse SDE.

2. When considering the general setting of VE-based methods, we propose the tangent-based unified framework, which contains reverse SDE and PFODE. Under this framework, we prove the first quantitative guarantee for SOTA VE-based models with reverse PFODE.

3. We show the power of our new forward process via synthetic experiments. For aggressive drift VESDE, we show it balances reverse beginning and discretization error. For conservative one, we show that it improves the quality of generated distribution without training.

## 2 Related Work

In the first two paragraphs of this section, we focus on works without the manifold hypothesis. In the last paragraph, we discuss the manifold hypothesis.

**Analyses for VP-based models.** De Bortoli et al. (2021) study VPSDE in TV distance and propose the first quantitative convergence guarantee with exponential dependence on problem parameters. Lee et al. (2022) achieve the first polynomial complexity with strong LSI assumption. Chen et al. (2023d) remove LSI assumption, assume the Lipschitz score and achieve polynomial complexity. Recently, Chen et al. (2023a) and Benton et al. (2023) further remove the Lipschitz score assumption, and Benton et al. (2023) achieve optimal dependence on $d$. When considering PFODE, Chen et al. (2023e) propose the first quantitative guarantee with exponential dependence on score Lipschitz constant. Chen et al. (2023c) achieve polynomial complexity by introducing a corrector component.

**Analyses for VE-based models.** When considering VESDE, most works focus on constant $\beta_t$ and reverse SDE setting. De Bortoli et al. (2021) provide the first quantitative convergence guarantee with exponential dependence on problem parameters. Lee et al. (2022) analyze a constant diffusion coefficient VESDE and achieve polynomial sample complexity. However, the results of Lee et al. (2022) relies heavily on the LSI assumption, which is unrealistic (Remark 1). When considering the reverse PFODE and unbounded $\beta_t$, Chen et al. (2023e) only consider the discretization error and provide a quantitative convergence guarantee. Furthermore, as discussed in Section 6.2, their results introduce additional $\exp(T)$ compared to ours.

**Analyses for diffusion models under the manifold hypothesis.** There are other line works (Pidstrigach, 2022; Bortoli, 2022; Lee et al., 2023; Chen et al., 2023d;b) that consider the manifold hypothesis. Pidstrigach (2022) is the first work considering the guarantee of VESDE, VPSDE, and CLD in the continuous process. Bortoli (2022) study VPSDE with reverse SDE, and it is the most relevant work to our unified framework. However, as discussed in Section 1, we need a refined analysis of the tangent process for VESDE with reverser PFODE to avoid $\exp(T)$. Chen et al. (2023d) and Lee et al. (2023) also analyze VPSDE and achieve the polynomial complexity. Chen et al. (2023b) study the score learning process and distribution recovery on the low dimensional data.

## 3   THE VARIANCE EXPLODING (VE) SDE FOR DIFFUSION MODELS

In this section, we introduce the basic knowledge of VE-based diffusion models from the perspective of SDE and discuss the assumption on the balance coefficient $\tau$ and diffusion coefficient $\beta_t$.

### 3.1   THE FORWARD PROCESS

First, we denote by $q_0$ the data distribution, $\mathbf{X}_0 \sim q_0 \in \mathbb{R}^d$, and $\{\beta_t\}_{t \in [0,T]}$ a non-decreasing sequence in $[0, T]$. Then, we define the forward process:

$$\mathrm{d}\mathbf{X}_t = -\frac{1}{\tau}\beta_t \mathbf{X}_t \, \mathrm{d}t + \sqrt{2\beta_t} \, \mathrm{d}\mathbf{B}_t, \quad \mathbf{X}_0 \sim q_0 \,, \tag{1}$$

where $(\mathbf{B}_t)_{t \geq 0}$ is a $d$-dimensional Brownian motion, and $\tau \in [T, T^2]$ is the coefficient to balance the drift and diffusion term. We also define $q_t^\tau$ as the marginal distribution of the forward OU process at time $t$. For well-defined $\beta_t$ (**Assumption 1**), we have that

$$\mathbf{X}_t = m_t \mathbf{X}_0 + \sigma_t Z, \; m_t = \exp\left[-\int_0^t \beta_s/\tau \, \mathrm{d}s\right], \; \sigma_t^2 = \tau\left(1 - \exp\left[-2\int_0^t \beta_s/\tau \, \mathrm{d}s\right]\right), \tag{2}$$

where $Z \sim \mathcal{N}(0, \mathbf{I})$. Later, we will discuss the choice of $\beta_t$, which depends on the type of reverse process and the choice of $\tau$. When the previous works consider VESDE, they usually consider the forward process without the drift term:

$$\mathrm{d}\mathbf{X}_t = \sqrt{\mathrm{d}\sigma_t^2/\mathrm{d}t} \, \mathrm{d}\mathbf{B}_t, \quad \mathbf{X}_0 \sim q_0 \,, \tag{3}$$

where $\sigma_t^2$ is a non-decreasing variance sequence. There are two common choices for VESDE. The first choice (Chen et al., 2023e) is $\sigma_t^2 = t$, whose marginal distribution $q_T$ is $\mathcal{N}(\mathbf{X}_0, T\mathbf{I})$. This choice is the setting that most theoretical works focus on (De Bortoli et al., 2021; Lee et al., 2022) and is similar to Eq. (1) with $\tau = T^2$ and $\beta_t = \frac{1}{2}, \forall t \in [T]$ since $q_T^\tau = \mathcal{N}(\exp(\frac{2}{T})\mathbf{X}_0, (1 - \exp(\frac{1}{T}))T^2\mathbf{I})$. The second SOTA choice (Karras et al., 2022; Song et al., 2023) is $\sigma_t^2 = t^2$, which corresponds to $\tau = T^2$, $\beta_t = t$ and $q_T^\tau = \mathcal{N}(e^{-1/2}\mathbb{E}[q_0], e^{-1}\mathrm{Cov}[q_0] + (1 - e^{-1})T^2\mathbf{I})$. We note that this $q_T^\tau$ still contains mean and variance information and almost identical to Eq. (3) with $\sigma_t^2 = t^2$. To support

our argumentation, we do simulation experiments and show that these two setting have similar trend (Fig. 1) . Hence, Eq. (1) is representative enough to represent current VESDE. Furthermore, the general forward SDE retaining the drift term will lead to a series of VESDE, which is helpful in choosing the tractable Gaussian distribution $q_\infty^\tau$ and enjoys a greater sample complexity (Section 5).

## 3.2 THE REVERSE PROCESS

Reversing the forward SDE, we obtain the reverse process $(\mathbf{Y}_t)_{t \in [0,T]} = (\mathbf{X}_{T-t})_{t \in [0,T]}$:

$$\mathrm{d}\mathbf{Y}_t = \beta_{T-t} \left\{ \mathbf{Y}_t/\tau + (1+\eta^2)\nabla \log q_{T-t}(\mathbf{Y}_t) \right\} \mathrm{d}t + \eta\sqrt{2\beta_{T-t}}\,\mathrm{d}\mathbf{B}_t\,, \qquad (4)$$

where now $(\mathbf{B}_t)_{t \geq 0}$ is the reversed Brownian Motion and $\eta \in [0,1]$. When $\eta = 1$, the reverse process corresponds to reverse SDE. When $\eta = 0$, the process corresponds to PFODE. Since $\nabla \log q_t$ can not be computed exactly, we need to use a score function $\mathbf{s}(t, \cdot)$ to approximate them. Then, we introduce the continuous-time reverse process $(\widehat{\mathbf{Y}}_t)_{t \in [0,T]}$ with approximated score :

$$\mathrm{d}\widehat{\mathbf{Y}}_t = \beta_{T-t}\{\widehat{\mathbf{Y}}_t/\tau + (1+\eta^2)\mathbf{s}(T-t, \widehat{\mathbf{Y}}_t)\}\mathrm{d}t + \eta\sqrt{2\beta_{T-t}}\,\mathrm{d}\mathbf{B}_t, \quad \widehat{\mathbf{Y}}_0 \sim q_\infty^\tau\,, \qquad (5)$$

where $q_\infty^\tau$ is the reverse beginning distribution, which always is a tractable Gaussian distribution. In this work, similar to Karras et al. (2022) and Song et al. (2023), we choose $q_\infty^\tau = \mathcal{N}(0, \sigma_T^2\mathbf{I})$. The last step is to discrete the continuous process. We define $\{\gamma_k\}_{k \in \{0,...,K\}}$ as the stepsize and $t_{k+1} = \sum_{j=0}^k \gamma_j$. In this work, we adapt the early stopping technique $t_K = T - \delta$, which has been widely used in practice (Ho et al., 2020; Kim et al., 2021; Karras et al., 2022). In this work, we consider the exponential integrator (EI) discretization (Zhang and Chen, 2022), which freezes the score function at time $t_k$ and defines the new SDE for small interval $t \in [t_k, t_{k+1}]$:

$$\mathrm{d}\widetilde{\mathbf{Y}}_t = \beta_{T-t}\{\widetilde{\mathbf{Y}}_t/\tau + (1+\eta^2)\mathbf{s}(T-t_k, \widetilde{\mathbf{Y}}_t)\}\mathrm{d}t + \eta\sqrt{2\beta_{T-t}}\,\mathrm{d}\mathbf{B}_t, \quad t \in [t_k, t_{k+1}]\,. \qquad (6)$$

Compared to the Euler–Maruyama (EM) discretization, EI has better experimental performance, and the results for EI can transfer to EM discretization (Bortoli, 2022). In this work, we consider $\beta_t$ can increase rapidly, for example, $\beta_t = t^2$, instead of a constant (Chen et al., 2023d) or in a small interval $[1/\bar{\beta}, \bar{\beta}]$ (Bortoli, 2022). Hence, we make the following assumption on $\beta_t$.

**Assumption 1.** *Define $t \mapsto \beta_t$ as a continuous, non-decreasing sequence. For any $\tau \in [T, T^2]$, there exists constants $\bar{\beta}$ and $G$, which are independent of $t$, such that for any $t \in [0, T]$: (1) if $\eta = 1$, then $1/\bar{\beta} \leq \beta_t \leq \max\{\bar{\beta}, t^2\}$; (2) if $\eta \in [0, 1)$, then $1/\bar{\beta} \leq \beta_t \leq \max\{\bar{\beta}, t\}$ and $\int_0^T \beta_t/\tau\,\mathrm{d}t \leq G$.*

This assumption rules out cases where $\beta_t$ grows too fast, such as $e^t$. We emphasize that $\beta_t$ grows slower than $t$ in the real world and satisfies $\int_0^T \beta_t/\tau\,\mathrm{d}t \leq G$ (Song et al., 2020b; Karras et al., 2022). However, our assumption is more general since $\beta_t$ depends on $\tau$ instead of at most linearly. For example, when $\eta = 1$ and $\tau = T^2$, we can choose $\beta_t = t^2$, which has the same order compared to $\tau$.

**Notations.** For $x \in \mathbb{R}^d$ and $A \in \mathbb{R}^{d \times d}$, we denote by $\|x\|$ and $\|A\|$ the Euclidean norm for vector and the spectral norm for matrix. We denote by $q_0 P_T$ the distribution of $\mathbf{X}_T$, $Q_{t_K}^{q_\infty^\tau}$ the distribution of $\mathbf{Y}_{t_K}$, $R_K^{q_\infty^\tau}$ the distribution of $\widetilde{\mathbf{Y}}_{t_K}$ and $Q_{t_K}^{q_0 P_T}$ the distribution which does forward process, then does reverse process (Eq. (4)). We denote by $W_1$ and $W_2$ the Wasserstein distance of order one and two.

## 4 THE FASTER FORWARD CONVERGENCE GUARANTEE FOR THE VESDE

Since the previous VESDE (Eq. (3)) does not converge to a stationary distribution, we usually choose a normal distribution $\mathcal{N}(\bar{m}_T, V_T)$ to approximate $q_T$. Pidstrigach (2022) show that the optimal solution is $\bar{m}_T = \mathbb{E}[q_0]$ and $V_T = \mathrm{Cov}[q_0] + \sigma_T^2\mathbf{I}$, which leads $1/\sigma_T^2$ forward convergence rate. To obtain a faster forward convergence rate, we introduce a new process, which allows $\mathbb{E}[q_0]$ and $\mathrm{Cov}[q_0]$ decay, as well as the variance explodes. In particular, we have the following lemma.

**Lemma 1.** *The minimization problem $\min_{\bar{m}_t, V_t} KL(q_t \mid \mathcal{N}(\bar{m}_t, V_t))$ is minimized by $\bar{m}_t = m_t\mathbb{E}[q_0]$ and $V_t = m_t^2\mathrm{Cov}[q_0] + \sigma_t^2\mathbf{I}$, where $m_t$ and $\sigma_t$ defined in Eq. (2).*

Lemma 1 is a general version compare to Pidstrigach (2022) since the existence of variable $m_t$ instead of constant one. Theorem 1 shows that variable $m_t$ allows a faster forward convergence rate. Before introducing the results, we introduce the manifold hypothesis on the data distribution.

**Assumption 2.** *$q_0$ is supported on a compact set $\mathcal{M}$ and $0 \in \mathcal{M}$.*

We denote $R$ the diameter of the manifold by $R = \sup\{\|x - y\| : x, y \in \mathcal{M}\}$ and assume $R > 1$. The manifold hypothesis is supported by much empirical evidence (Bengio et al., 2013; Fefferman et al., 2016; Pope et al., 2021) and is naturally satisfied by the image datasets since each channel of images is bounded. Furthermore, different from the Lipschitz score assumption, this assumption allows the expansion phenomenon of the score Kim et al. (2021) and is studied in the VPSDE setting. As shown in Bortoli (2022), this assumption also encompasses distributions which admit a continuous density on a lower dimensional manifold, and $R$ contains the dimension information. For example, considering a hypercube $\mathcal{M} = [-1/2, 1/2]^p$ with $p \leq d$, the $R = \sqrt{p}$, corresponding to the latent dimension $p$. Then, we obtain the following guarantee for the new forward process.

**Theorem 1.** *Let $q_T$ be the marginal distribution of the forward process, and $q_\infty^\tau = \mathcal{N}(0, \sigma_T^2 \mathbf{I})$ be the reverse beginning distribution. With $m_T, \sigma_T$ defined in Eq. (2), we have*

$$\|q_T - q_\infty^\tau\|_{TV} \leq \sqrt{m_T} \bar{D}/\sigma_T \,,$$

*where $\bar{D} = d|c| + \mathbb{E}[q_0] + R$ and $c$ is the eigenvalue of $Cov[q_0]$ with the largest absolute value.*

Recall that $m_T = \exp[-\int_0^T \beta_t/\tau \, \mathrm{d}t]$, the previous VESDE (Song et al., 2020b; Karras et al., 2022; Lee et al., 2022) choose a conservative $\beta_t$ satisfies $\int_0^T \beta_t/\tau \, \mathrm{d}t \leq G$. However, if we allow an aggressive $\beta_t$, the forward process will have a faster convergence rate. To illustrate the accelerated forward process, we use $\tau = T^2$ as an example and discuss the influence of different $\beta_t = t^{\alpha_1}, \alpha_1 \in [1, 2]$ under large enough $T$. Due to the definition of $\sigma_T$, $\sigma_T \approx T$, and the forward convergence rate mainly depends on $\sqrt{m_T}$. When $\alpha_1 = 1$ is conservative, $m_T$ is a constant, and the convergence rate is $1/T$. When $\alpha_1 = 1 + \ln(2r \ln(T))/\ln(T)$ is slightly aggressive, the convergence rate is $1/T^{r+1}$ for $r > 0$. When $\alpha_1 \geq 1 + \ln(T - \ln(T))/\ln(T)$ is aggressive, the forward convergence rate is faster than $\exp(-T)$. When $\alpha_1 = 2$ is the most aggressive choice, the convergence rate is $\exp(-T)/T$. In our analysis, whether $\beta_t$ can be aggressive in our unified framework depends on the form of the reverse process, as the discussion in Section 6. In the following section, we show that when choosing aggressive $\beta_t$ (reverse SDE setting), the new process balances the reverse beginning, discretization, and approximated score errors and achieves the first polynomial sample complexity for VE-based models under the manifold hypothesis.

## 5 THE POLYNOMIAL COMPLEXITY FOR VESDE WITH REVERSE SDE

In this section, we first pay attention to VESDE with reverse SDE to show the power of our new forward process and aggressive $\beta_t$. In this section, we assume an uniform $L_2$-accuracy assumption on scores, which is exactly the same compared to Chen et al. (2023d); Benton et al. (2023).

**Assumption 3** (Approximated score). *For all $k = 1, \ldots, K, \mathbb{E}_{q_{t_k}} \left[ \|s_{t_k} - \nabla \ln q_{t_k}\|^2 \right] \leq \epsilon_{score}^2.$*

We show that introducing aggressive $\beta_t$ only slightly affects the discretization error (additional logarithmic factors) and significantly benefits in balancing reverse beginning and discretization errors. Hence, we can obtain a polynomial sample complexity for VE-based models with unbounded $\beta_t$.

**Corollary 1.** *Assume **Assumption 1**, 2, 3. Let $\gamma_K = \delta$, $\bar{\gamma}_K = argmax_{k \in \{0, \ldots, K-1\}} \gamma_k$, $\tau = T^2$ and $\beta_t = t^2$. Then, $TV\left(R_K^{q_\infty^\tau}, q_0\right)$ is bounded by*

$$TV\left(R_K^{q_\infty^\tau}, q_0\right) \leq \frac{\bar{D} \exp(-T/2)}{T} + \frac{R^2 \sqrt{d}}{\delta^6} \sqrt{\bar{\gamma}_K T^5} + \epsilon_{score} \sqrt{T^3} \,,$$

*where $\bar{D} = d|c| + \mathbb{E}[q_0] + R$. Furthermore, by choosing $\delta \leq \frac{\epsilon_{W_2}^{2/3}}{(d + R\sqrt{d})^{1/3}}$, $T \geq 2 \ln \frac{\bar{D}}{\epsilon_{TV}}$, maximum stepsize $\bar{\gamma}_K \leq \delta^{12} \epsilon_{TV}^2 \ln^5(\bar{D}/\epsilon_{TV})/R^4 d$ and assuming $\epsilon_{score} \leq \widetilde{O}(\epsilon_{TV})$, the output of $R_K^{q_\infty^\tau}$ is $(\epsilon_{TV} + \epsilon_{score})$ close to $q_\delta$, which is $\epsilon_{W_2}$ close to $q_0$, with sample complexity (hiding logarithmic factors) is*

$$K \leq \widetilde{O}\left(\frac{dR^4(d + R\sqrt{d})^4}{\epsilon_{W_2}^8 \epsilon_{TV}^2}\right) \,.$$

*For choice $\beta_t = t$ and $\tau = T$, by choosing $\delta \leq \frac{\epsilon_{W_2}}{(d+R\sqrt{d})^{1/2}}$ and $\bar{\gamma}_K \leq \frac{\delta^8 \epsilon_{TV}^2 \ln^3(\bar{D}/\epsilon_{TV})}{R^4 d}$, we obtain the same sample complexity.*

First, we discuss the power of our new process and aggressive coefficient under the setting $\tau = T^2$, which is closer to the SOTA setting. Then, we discuss the results of $\tau = T$ and compare them to existing polynomial sample complexity results in Remark 1. For $\tau = T^2$, the above results show that our new process with aggressive $\beta_t = t^2$ balances the reverse beginning, discretization scheme and approximated score errors. If choosing a conservative $\beta_t = t$, term $\exp(-T/2)$ will be removed. Then, the guarantee has the form $1/T + \sqrt{\bar{\gamma}_K T^5}/\delta^6 + \epsilon_{\text{score}}\sqrt{T^3}$, which means if $T \geq 1/\epsilon_{\text{TV}}$, then $\epsilon_{\text{score}}\sqrt{T^3} \geq \epsilon_{\text{score}}/\sqrt{\epsilon_{\text{TV}}^3}$. Hence, it is hard to achieve non-asymptotic results for conservative $\beta_t$. However, by choosing aggressive $\beta_t$, $T$ becomes the logarithmic factor, and these error sources are balanced. In Section 7.1, we do experiments on 2-D Gaussian to support our above augmentation.

**Remark 1.** *Lee et al. (2022) consider pure VESDE (Eq. (3)) with $\sigma_t^2 = t$ and reverse SDE under the LSI assumption with parameter $C_{LS}$. The LSI assumption does not allow the presence of substantial non-convexity and is far away from the multi-modal real-world distribution. Furthermore, they use unrealistic assumption $\epsilon_{score} \leq 1/(C_{LS} + T)$ to avoid the effect of the approximated score, which is stronger than* **Assumption 3**. *Under the above strong assumption on data and approximated score function, Lee et al. (2022) achieve the polynomial sample complexity $\tilde{O}(L^2 d(d|c| + R)^2/\epsilon_{TV}^4)$. Under the manifold hypothesis and choosing $L = R^2 d^2/\epsilon_{W_2}^4$, the sample complexity is $\tilde{O}(R^4 d^5(d|c| + R)^2/\epsilon_{W_2}^8 \epsilon_{TV}^4)$, which worse than Corollary 1.*

## 6 THE UNIFIED ANALYSIS FRAMEWORK FOR VE-BASED METHODS

In this section, we go beyond the reverse SDE setting and introduce the unified analysis framework for VESDE with reverse SDE and PFODE. In Section 6.1, we show the convergence results in our unified framework. In Section 6.2, we introduce the detail of our tangent-based unified framework and discuss the variance exploding property of VESDE, which allows analyzing reverse PFODE.

### 6.1 THE CONVERGENCE GUARANTEE FOR VESDE

In this part, we consider the reverse beginning and discretization errors and assume an accurate score. This setting is similar to Chen et al. (2023e), which mainly considers the reverse PFODE. It is a meaningful step to analyze VESDE with reverse SDE and PFODE in a unified framework and show the property of VESDE instead of the property of reverse process.

**Theorem 2.** *Assume* **Assumption 1** *and 2,* $\gamma_k \sup_{v \in [T-t_{k+1}, T-t_k]} \beta_v/\sigma_v^2 \leq 1/28$ *for* $\forall k \in \{0, ..., K-1\}$, *and* $\delta \leq 1/32$. *Let* $\bar{\gamma}_K = argmax_{k \in \{0,...,K-1\}} \gamma_k$ *and* $\gamma_K = \delta$. *Then, for* $\forall \tau \in [T, T^2]$, *we have the following convergence guarantee.*

*(1) If $\eta = 1$ (the reverse SDE), choosing an aggressive $\beta_t = t^2$, we have*

$$W_1\left(R_K^{q_\infty^\tau}, q_0\right) \leq C_1(\tau) T \exp\left[\frac{R^2}{2}(\frac{\bar{\beta}}{\delta^3} + \frac{1}{\tau})\right] [\kappa_1^2(\tau)(\frac{\bar{\beta}}{\delta^3} + \frac{1}{\tau})\bar{\gamma}_K^{1/2} + \kappa_1^2(\tau)]\bar{\gamma}_K^{1/2}$$

$$+ \exp\left[\frac{R^2}{2}(\frac{\bar{\beta}}{\delta^3} + \frac{1}{\tau})\right] \frac{\bar{D}\exp(-T/2)}{\sqrt{\tau}} + 2(\frac{R}{\tau} + \sqrt{d})\sqrt{\delta},$$

*where $C_1(\tau)$ is linear in $\tau^2$, $\kappa_1(\tau) = \max\{\bar{\beta}, T^2\}(1/\tau + \bar{\beta}/\delta^3)$.*

*(2) If $\eta = 0$ (the reverse PFODE), choosing a conservative $\beta_t$ satisfies* **Assumption 1**, *we have*

$$W_1\left(R_K^{q_\infty^\tau}, q_0\right) \leq C_2(\tau) T \exp\left[\frac{R^2}{2}(\frac{\bar{\beta}}{\delta^2} + \frac{1}{\tau}) + \frac{1}{2}\right] [\kappa_2^2(\tau)(\frac{\bar{\beta}}{\delta^2} + \frac{1}{\tau})\bar{\gamma}_K^{1/2} + \kappa_2^2(\tau)]\bar{\gamma}_K^{1/2}$$

$$+ \exp\left[\frac{R^2}{2}(\frac{\bar{\beta}}{\delta^2} + \frac{1}{\tau})\right] \frac{\bar{D}}{\sqrt{\tau}} + 2(\frac{R}{\tau} + \sqrt{d})\sqrt{\delta},$$

*where $C_2(\tau)$ is linear in $\tau^2$, $\kappa_2(\tau) = \max\{\bar{\beta}, T\}(1/\tau + \bar{\beta}/\delta^2)$.*

Theorem 2 proves the first quantitative guarantee for VE-based models with reverse PFODE in the unified tangent-based method (Section 6.2). We also show that this framework can deal with

reverse SDE. As shown in Theorem 2, for different reverse processes, our unified framework chooses different $\beta_t$ to achieve a quantitative convergence guarantee, which shows the power of our framework. Correspondingly, the Girsanov-based method (Chen et al., 2023d;a; Benton et al., 2023) can not obtain the guarantee of reverse PFODE since the diffusion term for reverse process is not well-defined.

For the reverse PFODE, Chen et al. (2023e) employ the Restoration-Degradation framework to analyze the VESDE. Since their result has an exponential dependence on the Lipschitz constant of the score, their results also have exponential dependence on $R$ and $\delta$. Furthermore, their results have exponential dependence on the growth rate of $\beta_t$ ($g_{\max}$ in (Chen et al., 2023e)), which corresponds to $\tau$ of VESDE. However, our dependence on $\tau$ appears in the polynomial term. Hence, our framework is a suitable unified framework for VE-based methods. Furthermore, we emphasize that our tangent-based unified framework is not a simple extension of Bortoli (2022). We carefully control the tangent process according to the variance exploding property of VESDE to avoid $\exp(T)$ term when considering PFODE, as discussed in Section 6.2.

**Corollary 2.** *Assume **Assumption 1** and **2**. Let $\epsilon \in (0, 1/32), \tau \in [T, T^2], \gamma_K = \delta = \epsilon^2, \bar{\gamma}_K = argmax_{k \in \{0, \dots, K-1\}} \gamma_k$,*

*(1) if $\eta = 1$, with an aggressive $\beta_t = t^2$, $T \geq \bar{\beta}(R^2 + 1)/\epsilon^6$, $\bar{\gamma}_K \leq \exp(-T)/(T^{\frac{20}{3}} \tau^4 C_1^2(\tau))$;*

*(2) if $\eta = 0$, with a conservative $\beta_t$ (**Assumption 1**), $\tau \geq \exp(\frac{R^2 \bar{\beta}}{\epsilon^4})/\epsilon^2$ and $\bar{\gamma}_K \leq \frac{1}{\tau T^6 \ln^2(T) C_2^2(\tau)}$:*

$$W_1\left(R_K^{q_\infty^\tau}, q_0\right) \leq (\bar{D} + 2R^2 + 2\sqrt{d})\epsilon,$$

*where $\bar{D}$ is defined in Theorem 1, and $\max\{C_1(\tau), C_2(\tau)\} \leq (16 + \bar{\beta}^{\frac{3}{2}} + 1)(2 + R^2)(12R + 4\tau^2\sqrt{d})$.*

We note that Theorem 2 has exponential dependence on $R$ and $\delta$, which is introduced by the tangent process. Similar to Bortoli (2022), if we assume the Hessian $\|\nabla^2 \log q_t(x_t)\| \leq \Gamma/\sigma_t^2$, we obtain a better control on the tangent process and replace the exponential dependence on $\delta$ by a polynomial dependence on $\delta$ and exponential dependence on $\Gamma$ when considering reverse PFODE.

**Corollary 3.** *Assume **Assumption 1**, **Assumption 2** and $\|\nabla^2 \log q_t(x_t)\| \leq \Gamma/\sigma_t^2$. Let $\eta = 0$ (reverse PFODE), $\epsilon \in (0, 1/32), \tau = T^2$, $\beta_t = t$, $\bar{\gamma}_K = argmax_{k \in \{0, \dots, K-1\}} \gamma_k, \gamma_K = \delta$, we have*

$$W_1\left(R_K^{q_\infty^\tau}, q_0\right) \leq C_2(\tau) T \frac{\bar{\beta}^{\frac{\Gamma}{2}}}{\delta^\Gamma} \exp\left[\frac{\Gamma + 2}{2}\right] [\kappa_2^2(\tau)(\frac{\bar{\beta}}{\delta^2} + \frac{1}{\tau})\bar{\gamma}_K^{1/2} + \kappa_2^2(\tau)]\bar{\gamma}_K^{1/2}$$

$$+ \frac{\bar{\beta}^{\frac{\Gamma}{2}}}{\delta^\Gamma} \exp\left[\frac{\Gamma + 2}{2}\right] \frac{\bar{D}}{\sqrt{\tau}} + 2(\frac{R}{\tau} + \sqrt{d})\sqrt{\delta},$$

*where $C_2(\tau)$ is linear in $\tau^2$, $\kappa_2(\tau) = \max\{\bar{\beta}, T\}\left(\frac{1}{\tau} + \frac{\bar{\beta}}{\delta^2}\right)$.*

Though the $\Gamma/\sigma_\delta^2$ bound is stronger than $(1 + R^2)/\sigma_\delta^4$ in Theorem 2 and Chen et al. (2023d), there are many special cases such as hypercube $\mathcal{M} = [-1/2, 1/2]^p$ with $p \leq d$ satisfy this assumption.

**Remark 2.** *Since the reverse PFODE setting does not involve the aggressive $\beta_t = t^2$, our analysis still holds if we consider original VESDE Eq. (3) with $\sigma_t^2 = t^2$ and reverse PFODE, which means our analysis can explain the results in (Karras et al., 2022). For consistency, we use Eq. (1) in this work.*

**Remark 3.** *In this section, similar to Chen et al. (2023e), we assume an accurate score as the first step. When considering the approximated score, our guarantee has an additional $\epsilon_{score} T$ term. As discussed in Remark 1, we can eliminate the effect of $\epsilon_{score}$ by adding strong assumption. However, if assuming **Assumption 3**, since $T$ in Corollary 2 is not a logarithmic factor, we can not ignore it. One future work is considering an approximated score in PFODE and achieving a polynomial complexity.*

### 6.2 THE DICUSSION ON THE UNIFIED FRAMEWORK

In this section, we introduce the unified tangent-based framework for reverse SDE and PFODE and discuss key steps to achieve the quantitative guarantee for PFODE. Firstly, we decompose the goal:

$$W_1\left(R_K^{q_\infty^\tau}, q_0\right) \leq W_1\left(R_K^{q_\infty^\tau}, Q_{t_K}^{q_\infty^\tau}\right) + W_1\left(Q_{t_K}^{q_\infty^\tau}, Q_{t_K}^{q_0 P_T}\right) + W_1\left(Q_{t_K}^{q_0 P_T}, q_0\right).$$

These terms correspond to the discretization scheme, reverse beginning distribution, and the early stopping parameter $\delta$. We focus on most difficult discretization term and first recall the stochastic flow of the reverse process for any $x \in \mathbb{R}^d$ and $s, t \in [0, T]$ with $t \geq s$:

$$\mathrm{d}\mathbf{Y}_{s,t}^x = \beta_{T-t}\left\{\mathbf{Y}_{s,t}^x/\tau + \left(1 + \eta^2\right)\nabla\log q_{T-t}\left(\mathbf{Y}_{s,t}^x\right)\right\}\mathrm{d}t + \eta\sqrt{2\beta_{T-t}}\mathrm{d}\mathbf{B}_t, \qquad \mathbf{Y}_{s,s}^x = x\,,$$

and the corresponding tangent process

$$\mathrm{d}\nabla\mathbf{Y}_{s,t}^x = \beta_{T-t}\left\{\mathbf{I}/\tau + \left(1 + \eta^2\right)\nabla^2\log q_{T-t}(\mathbf{Y}_{s,t}^x)\right\}\nabla\mathbf{Y}_{s,t}^x\mathrm{d}t, \qquad \nabla\mathbf{Y}_{s,s}^x = \mathbf{I}\,, \qquad (7)$$

which is used to control the discretization error in Bortoli (2022). Then, the discretization error is bounded by time and space discretization error for a small interval $[t_k, t_{k+1}]$ and the tangent process $\left\|\nabla\mathbf{Y}_{s,t_K}^x\right\|$ for $\forall s \in [0, t_K]$. For the first two terms, we control the Lipshctiz constant $\left\|\nabla^2\log q_t\left(x_t\right)\right\|$ and score perturbation $\left\|\partial_t\nabla\log q_t\left(x_t\right)\right\|$ at time $t \in [t_k, t_{k+1}]$. For the key $\left\|\nabla\mathbf{Y}_{s,t_K}^x\right\|$, we consider the reverse SDE and PFODE simultaneously and propose a general version of Bortoli (2022). Since the bound of tangent process depend on $\sigma_{T-t_K}^{-2}$, which corresponding to $\beta_t$, we introduce an indicator $i \in \{1, 2\}$ for $\sigma_{T-t_K}$. We use $\tau = T^2$ as an example. When $\beta_t = t^2$ is aggressive, $i = 1$, $\eta = 1$ and $\sigma_{T-t_K}^{-2}(i = 1) \leq 1/\tau + \bar{\beta}/\delta^3$. When $\beta_t = t$ is conservative, $i = 2$, $\eta \in [0, 1)$ and $\sigma_{T-t_K}^{-2}(i = 2) \leq 1/\tau + \bar{\beta}/\delta^2$. Then, we obtain the following guarantee for the tangent process.

**Lemma 2.** *Assume* **Assumption 1** *and* **2**. *Then, for* $\forall s \in [0, t_K], x \in \mathbb{R}^d$, *and* $i \in \{1, 2\}$, *we have*

$$\|\nabla\mathbf{Y}_{s,t_K,i}^x\| \leq \exp\left[\frac{R^2}{2\sigma_{T-t_K}^2(i)} + \frac{(1 - \eta^2)}{2}\int_0^{t_K}\frac{\beta_{T-u}}{\tau}\mathrm{d}u\right]\,.$$

*Furthermore, if assuming* $\left\|\nabla^2\log q_t\left(x_t\right)\right\| \leq \Gamma/\sigma_t^2$, *we have that*

$$\|\nabla\mathbf{Y}_{s,t_K,i}^x\| \leq \sigma_{T-t_K}^{-(1+\eta^2)\Gamma}(i)\exp\left[\left(\left(1 + \eta^2\right)\Gamma + 2\right)\int_0^{t_K}\frac{\beta_{T-u}}{\tau}\mathrm{d}u\right]\,.$$

We emphasize that the general bound for the tangent process is the key to achieving the guarantee for VESDE with the reverse ODE. Recall that in the original lemma for the tangent processes, since $\tau$ is independent of $T$ and $\beta_t$ is bounded in a small interval $[1/\bar{\beta}, \bar{\beta}]$, $\int_0^{t_K}\beta_{T-u}/\tau\mathrm{d}u = \Theta(T)$, which means there is an additional $\exp(T)$ when considering VPSDE with revere PFODE. However, our tangent-based lemma makes use of the variance exploding property of VESDE to guarantee that $\int_0^T\beta_t/\tau\mathrm{d}t \leq G$ with a conservative $\beta_t = t$ when considering reverse PFODE. When considering reverse SDE ($\eta = 1$), we can choose aggressive $\beta_t = t^2$ since the choice of $\beta_t$ does not affect the bound of the tangent process. Then, we control the discretization error for $\eta \in [0, 1]$. For the remaining two term, we know that the early stopping terms is smaller than $2(R/\tau + \sqrt{d})\sqrt{\delta}$ and

$$W_1\left(Q_{t_K}^{q_\infty^\tau}, Q_{t_K}^{q_0 P_T}\right) \leq \frac{\sqrt{m_T}\bar{D}}{\sigma_T}\exp\left[\frac{R^2}{2\sigma_{T-t_K}^2(i)} + \frac{(1 - \eta^2)}{2}\int_0^{t_K}\frac{\beta_{T-u}}{\tau}\mathrm{d}u\right]\,,$$

The reason why an exponential dependence in reverse beginning term is that we can not use the data processing inequality in Wasserstein distance. One future work is introducing the short regularization technique (Chen et al., 2023c) and suitable corrector to remove this exponential dependence.

# 7 SYNTHETIC EXPERIMENTS

In this section, we do some synthetic experiments to show the power of the new forward process. In Section 7.1, we show that with aggressive $\beta_t$, the new process achieves good balance in different error terms. Furthermore, we consider the approximated score and show that the conservative drift VESDE can improve the quality of the generated distribution without training.

## 7.1 THE POWER OF AGGRESSIVE VESDE IN BALANCING DIFFERENT ERROR SOURCES

In this section, we do experiments on 2-D Gaussian to show that the drift VESDE with aggressive $\beta_t$ has power in balancing the reverse beginning and discretization errors. Since the ground truth score of the Gaussian can be directly calculated, we use the accurate score function to discuss the balance between the other two error terms. We show how to use approximated score in Section 7.2.

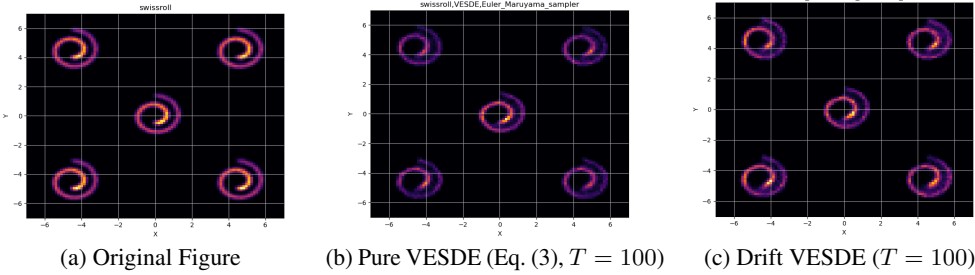

(a) Original Figure     (b) Pure VESDE (Eq. (3), $T = 100$)     (c) Drift VESDE ($T = 100$)

Figure 2: Experiment results of Swiss roll with Euler Maruyama Method (Reverse SDE)

As shown in Fig. 1, the process with aggressive $\beta_t = t^2$ achieves the best and second performance in EI and EM discretization, which support our theoretical result (Corollary 1). The third best process is conservative $\beta_t = t$ with the small drift term. The reason is that though it can not achieve a $\exp(-T)$ forward guarantee, it also has a constant decay on prior information, as shown in Section 3.1. This decay slightly reduces the effect of the reverse beginning error. The worst process is pure VESDE since it is hard to balance different error sources. Our experimental results also show that EI is better than EM discretization.

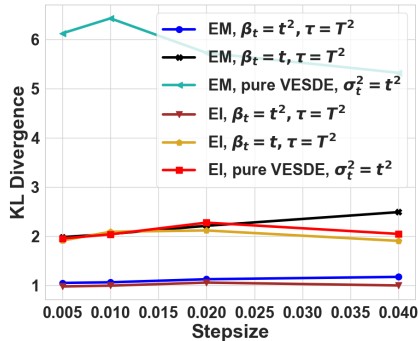

Figure 1: Experiment results of 2-D Gaussian

## 7.2 THE CONSERVATIVE DRIFT VESDE BENEFITS FROM PURE VESDE WITHOUT TRAINING

As shown in Fig. 1, the red and orange lines have similar trends. Hence, for conservative drift VESDE, which satisfies (2) of **Assumption 1**, we can directly use the models trained by pure VESDE to improve the quality of generated distribution. We confirm our intuition by training the model with pure VESDE (Eq. (3)) with $\sigma_t^2 = t$ and use the models to conservative drift VESDE with $\beta_t = 1$ and $\tau = T$. From the experimental results (Fig. 2), it is clear that pure VESDE has a low density on the Swiss roll except for the center one, which indicates pure VESDE can not deal with large dataset variance $\text{Cov}[q_0]$, as we discuss in Section 4. For conservative drift VESDE ($\beta_t = 1$ and $\tau = T$), as we discuss in the above section, it can reduce the influence of the dataset information. Fig. 2c support our augmentation and show that the density of the generated distribution is more uniform compared to pure VESDE, which means that the drift VESDE can deal with large dataset mean and variance.

There are more experiments on Swiss roll and 1D-GMM to explore different sampling methods (RK45, reverse PFODE) and different $T$. We refer to Appendix F for more details and discussion.

## 8 CONCLUSION

In this work, we analyze the VE-based models under the manifold hypothesis. Firstly, we propose a new forward VESDE process by introducing a small drift term, which enjoys a faster forward convergence rate than the Brownian Motion. Then, we show that with an aggressive $\beta_t$, the new process has the power to balance different error sources and achieve the first polynomial sample complexity for VE-based models with unbounded coefficient and reverse SDE.

After achieving the above results, we go beyond the reverse SDE and propose the tangent-based unified framework, which contains reverse SDE and PFODE. Under this framework, we make use of the variance exploding property of VESDE and achieve the first quantitative convergence guarantee for SOTA VE-based models with reverse PFODE. Finally, we do synthetic experiments to show the power of the new forward process.

**Future Work.** This work proposes the first unified framework for VE-based models with an accurate score. After that, we plan to consider the approximated score error and provide a polynomial sample complexity for the VE-based models with reverse PFODE under the manifold hypothesis.

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
