**Proof.** For simplicity, we denote the mean and covariance of $q_0$ by $a$ and $C$. We also define the optimize variable $n_t = \mathcal{N}\left(\bar{m}_t, C_t\right)$. We can directly compute the KL divergence $KL(q_t|n_t)$:

$$
\begin{aligned}
KL\left(q_t|n_t\right) &= -H\left(q_t\right) - \int \log\left(n_t(x)\right) q_t(x)\mathrm{d}x \\
&= -H\left(q_t\right) + \frac{d}{2}\log(2\pi) + \frac{1}{2}\log\left(\det\left(V_t\right)\right) + \frac{1}{2}\int (x - \bar{m}_t)^T V_t^{-1}(x - \bar{m}_t)q_t(x)\mathrm{d}x\,.
\end{aligned}
$$

For the last term, we directly compute

$$
\begin{aligned}
&\int \left(x - \bar{m}_t\right)^T V_t^{-1}\left(x - \bar{m}_t\right) p_t(x)\mathrm{d}x \\
=&\mathbb{E}\left[\left(X_t - \bar{m}_t\right)^T V_t^{-1}\left(X_t - \bar{m}_t\right)\right] = \mathbb{E}\left[\left(m_t X_0 + \sigma_t Z - \bar{m}_t\right)^T V_t^{-1}\left(m_t X_0 + \sigma_t Z - \bar{m}_t\right)\right] \\
=&\mathbb{E}\left[m_t^2\left(X_0 - a\right)^T V_t^{-1}\left(X_0 - a\right)\right] + \left(m_t a - \bar{m}_t\right)^T V_t^{-1}\left(m_t a - \bar{m}_t\right) + \sigma_t^2 \mathbb{E}\left[Z^T V_t^{-1} Z\right] \\
=&m_t^2 \operatorname{tr}\left(C V_t^{-1}\right) + \sigma_t^2 \operatorname{tr}\left(V_t^{-1}\right) + \left(m_t a - \bar{m}_t\right)^T V_t^{-1}\left(m_t a - \bar{m}_t\right)\,,
\end{aligned}
$$

where the second inequality follows that $X_t = m_t X_0 + \sigma_t Z$. It is clear that the optimal solution of $\bar{m}_t$ is $m_t a$. In the next step, we focus on the optimization problem for $V_t$:

$$
\begin{aligned}
L\left(V_t^{-1}\right) &= \log\left(\det\left(V_t\right)\right) + \operatorname{tr}\left(\left(m_t^2 C + \sigma_t^2 \mathbf{I}\right) V_t^{-1}\right) \\
&= -\log\left(\det\left(V_t^{-1}\right)\right) + \operatorname{tr}\left(\left(m_t^2 C + \sigma_t^2 \mathbf{I}\right) V_t^{-1}\right)\,.
\end{aligned}
$$

Since the above optimization is a convex optimization problem, we use the method similar to Pidstrigach (2022), we obtain that the optimal solution of $V_t$ is $m_t^2 C + \sigma_t^2 \mathbf{I}$. ∎

**Lemma 3.** *Let* $\bar{m}_t$ *and* $V_t$ *be the optimal mean and covariance operator from Lemma 1. Then*

$$
\begin{aligned}
KL\left(q_t|\mathcal{N}(\bar{m}_t, V_t)\right) &\leq \frac{1}{2}\log\left(\frac{\prod_{i=1}^d\left(m_t^2 c_i + \sigma_t^2\right)}{(\sigma_t^2)^d}\right) + \frac{R^2 m_t}{\sigma_t^2} \\
&\leq \frac{dm_t^2 c}{2\sigma_t^2} + \frac{R^2 m_t}{\sigma_t^2} + o(\frac{m_t^2 c}{\sigma_t^2})\,, \\
KL\left(\mathcal{N}(\bar{m}_t, V_t)|(\mathcal{N}(0, \sigma_t^2))\right) &\leq \frac{m_t^2 \sum_{i=1}^d c_i}{2\sigma_t^2} + \frac{m_t^2(\mathbb{E}[q_0])^2}{2\sigma_t^2} + \frac{1}{2}\log\left(\frac{(\sigma_t^2)^d}{\prod_{i=1}^d\left(m_t^2 c_i + \sigma_t^2\right)}\right) \\
&\leq \frac{m_t^2 \sum_{i=1}^d c_i}{2\sigma_t^2} + \frac{m_t^2(\mathbb{E}[q_0])^2}{2\sigma_t^2} + \frac{dm_t^2 c}{2\sigma_t^2} + o(\frac{m_t^2 c}{\sigma_t^2})\,,
\end{aligned}
$$

*where* $c_i$ *are the eigenvalues of* $\operatorname{Cov}\left[q_0\right]$, *and* $c$ *is the eigenvalue with the largest absolute value.*

**Proof.** For $t \geq 0$, we directly calculate the KL divergence for this term:

$$
\begin{aligned}
\mathrm{KL}\left(q_t \mid \mathcal{N}(\bar{m}_t, V_t)\right) &= -H\left(q_t\right) + \frac{1}{2}\log\left(\det\left(2\pi V_t\right)\right) + \frac{1}{2}\operatorname{tr}\left(\left(m_t^2 C + \sigma_t^2 \mathbf{I}\right) V_t^{-1}\right) \\
&= -H\left(q_t\right) + \frac{1}{2}\log\left(\det\left(2\pi V_t\right)\right) + \frac{d}{2} \\
&= -H\left(q_t\right) + \frac{d}{2}\log(2\pi) + \frac{1}{2}\log\left(\prod_{i=1}^d\left(m_t^2 c_i + \sigma_t^2\right)\right) + \frac{d}{2}\,,
\end{aligned}
$$

where $c_i$ are the eigenvalues of $\text{Cov}\,[q_0]$. Now, we only need to calculate $H(q_t)$:

$$-H\left(q_t\right) = \mathbb{E}_{X_t}\left[\log q_t\left(X_t\right)\right] = \mathbb{E}_{X_t}\left[\log\left(\mathbb{E}_{X_0}\left[(2\pi\sigma_t^2)^{-d/2}\exp\left(-\frac{1}{2\sigma_t^2}\left\|X_t - X_0\right\|^2\right)\right]\right)\right].$$

By **Assumption 2**, it is clear that

$$\exp\left(-\frac{1}{2\sigma_t^2}\left\|X_t - X_0\right\|^2\right) \leq \exp\left(-\frac{1}{2\sigma_t^2}\left(\left\|X_t\right\|^2 + 2\langle X_t, X_0\rangle\right)\right).$$

Then, we know that

$$\mathbb{E}\left[\log\left(\mathbb{E}_{X_0}\left[(2\pi\sigma_t^2)^{-d/2}\exp\left(-\frac{1}{2\sigma_t^2}\left\|X_t - X_0\right\|^2\right)\right]\right)\right]$$

$$\leq\mathbb{E}\left[\log\left((2\pi\sigma_t^2)^{-d/2}\right) - \frac{1}{2\sigma_t^2}\left(\left\|X_t\right\|^2 + 2\langle X_t, X_0\rangle\right)\right]$$

$$\leq -\frac{d}{2}\log(2\pi) - \frac{1}{2}\log\left((\sigma_t^2)^d\right) - \frac{1}{2\sigma_t^2}\mathbb{E}\left[\left\|X_t\right\|^2\right] + \frac{R^2 m_t}{\sigma_t^2}.$$

we also know that

$$\mathbb{E}\left[\left\|X_t\right\|^2\right] = m_t^2\mathbb{E}\left[\left\|X_0\right\|^2\right] + \sigma_t^2\mathbb{E}\left[\left\|Z\right\|^2\right] = \mathbb{E}\left[\left\|X_0\right\|^2\right] + t\mathbb{E}\left[\left\|Z\right\|^2\right] = \bar{m}_0^2 + V_0 + \sigma_t^2 d.$$

Finally, put these terms together, we have:

$$\text{KL}\left(q_t|\mathcal{N}(\bar{m}_t, V_t)\right) \leq \frac{1}{2}\log\left(\frac{\prod_{i=1}^d\left(m_t^2 c_i + \sigma_t^2\right)}{(\sigma_t^2)^d}\right) + \frac{R^2 m_t}{\sigma_t^2},$$

where $c_i$ are the eigenvalues of $\text{Cov}\,[q_0]$. Then by choosing the largest absolute value eigenvalue largest absolute value, we can use the Taylor expansion to obtain the first results of this lemma. For the second result of this lemma, we directly compute the KL divergence between $\mathcal{N}(\bar{m}_t, V_t)$ and $\mathcal{N}(0, \sigma_t^2)$ to obtain the final results. ∎

**Theorem 1.** *Let $q_T$ be the marginal distribution of the forward process, and $q_\infty^\tau = \mathcal{N}(0, \sigma_T^2\mathbf{I})$ be the reverse beginning distribution. With $m_T, \sigma_T$ defined in Eq. (2), we have*

$$\|q_T - q_\infty^\tau\|_{TV} \leq \sqrt{m_T}\bar{D}/\sigma_T,$$

*where $\bar{D} = d|c| + \mathbb{E}[q_0] + R$ and $c$ is the eigenvalue of $\text{Cov}[q_0]$ with the largest absolute value.*

**Proof.** We know that

$$\|q_T - q_\infty^\tau\|_{\text{TV}}$$
$$\leq \|q_T - \mathcal{N}(m_T\mathbb{E}[q_0], m_T^2\text{Cov}[q_0] + \sigma_T^2\mathbf{I})\|_{\text{TV}} + \|\mathcal{N}(m_T\mathbb{E}[q_0], m_T^2\text{Cov}[q_0] + \sigma_T^2\mathbf{I}) - q_\infty^\tau\|_{\text{TV}}.$$

By directly using the Pinsker's inequality and Lemma 3, we complete the proof. ∎

## B  THE PROOF OF THE POLYNOMIAL COMPLEXITY FOR REVERSE SDE

In this section, we prove Corollary 1. First, we recall the Girsanov's Theorem (Le Gall, 2016) used in Chen et al. (2023d):

**Lemma 4** (Girsanov's theorem)**.** *Let $P_T$ and $Q_T$ be two probability measures on path space $\mathcal{C}\left([0, T]; \mathbb{R}^d\right)$. Suppose that under $P_T$, the process $(X_t)_{t\in[0,T]}$ follows*

$$\mathrm{d}X_t = \tilde{b}_t\,\mathrm{d}t + \alpha_t\,\mathrm{d}\tilde{B}_t$$

*where $\tilde{B}$ is a $P_T$-Brownian motion, and under $Q_T$, the process $(X_t)_{t\in[0,T]}$ follows*

$$\mathrm{d}X_t = b_t\,\mathrm{d}t + \alpha_t\,\mathrm{d}B_t$$

*where $B$ is a $Q_T$-Brownian motion. We assume that for each $t > 0, \alpha_t$ is a $d \times d$ symmetric positive definite matrix. Then, provided that Novikov's condition holds,*

$$\mathbb{E}_{Q_T} \exp \left( \frac{1}{2} \int_0^T \left\| \alpha_t^{-1} \left( \tilde{b}_t - b_t \right) \right\|^2 \, \mathrm{d}t \right) < \infty,$$

*we have that*

$$\frac{\mathrm{d}P_T}{\mathrm{d}Q_T} = \exp \left( \int_0^T \alpha_t^{-1} \left( \tilde{b}_t - b_t \right) \mathrm{d}B_t - \frac{1}{2} \int_0^T \left\| \alpha_t^{-1} \left( \tilde{b}_t - b_t \right) \right\|^2 \, \mathrm{d}t \right).$$

If the Novikov's condition is satisfied, we apply the Girsanov theorem by choosing $P_T = R_K^{q_T^\tau}, Q_T = Q_{t_K}^{q_T^\tau}, \tilde{b}_t = \beta_{T-t} \left\{ \frac{1}{\tau} \widetilde{\mathbf{Y}}_t + 2\mathbf{s}(T - t_k, \widetilde{\mathbf{Y}}_t) \right\}$ (for $t \in [t_k, t_{k+1}]$), $b_t = \beta_{T-t} \left\{ \frac{1}{\tau} \mathbf{Y}_t + \left( 1 + \eta^2 \right) \nabla \log q_{T-t} (\mathbf{Y}_t) \right\}$, and $\alpha_t = \sqrt{2\beta_{T-t}} \mathbf{I}_d$.

Then, similar to Chen et al. (2023d), we have the following lemma.

**Lemma 5.** *Assuming that $R_K^{q_T^\tau}$ and $Q_{t_K}^{q_T^\tau}$ satisfy Novikov's condition, it holds that*

$$\mathrm{KL} \left( Q_{t_K}^{q_T^\tau} \| R_K^{q_T^\tau} \right)$$

$$= \mathbb{E}_{Q_{t_K}^{q_T^\tau}} \ln \frac{\mathrm{d}Q_{t_K}^{q_T^\tau}}{\mathrm{d}R_K^{q_T^\tau}} = \sum_{k=0}^{K-1} \mathbb{E}_{Q_{t_K}^{q_T^\tau}} \int_{t_k}^{t_{k+1}} 2\beta_{T-t} \left\| \mathbf{s} \left( T - t_k, \mathbf{Y}_{t_k} \right) - \nabla \ln q_{T-t} \left( \mathbf{Y}_t \right) \right\|^2 \, \mathrm{d}t.$$

Before using the Girsanov's Theorem, we need to check the Novikov's condition. The key proof of the Novikov's condition is Lemma 19 of Chen et al. (2023d). Since we assume the accurate score function in this work, this lemma need to control

$$\sup_{x^* \in \mathrm{B}(0,R), t^* \in [0, T-\delta]} 2\beta_{T-t^*} \left\| \nabla \ln q_{T-t^*} \left( x^* \right) \right\| =: B < \infty.$$

As we shown in Lemma 13, we know that with the early stopping parameter $\delta$, $\left\| \nabla \ln q_{T-t^*} \left( x^* \right) \right\|$ is controlled. By using **Assumption 2**, we know that $\frac{1}{\beta_{T-t^*}} \leq \bar{\beta}$. Finally, with similar process to Chen et al. (2023d), we can proof that the Novikov's condition is satisfied. The following lemma show the discretization error for VESDE with reverse SDE.

**Lemma 6** (Discretization). *Suppose that **Assumption 2** and **Assumption 3** holds. Let $\bar{\gamma}_K = \mathrm{argmax}_{k \in \{0, \dots, K-1\}} \gamma_k, \gamma_K = \delta,$*

*(1) If $\tau = T^2$ and $\beta_t = t^2$, then with $Q_{t_K}^{q_T^\tau}$ and $R_K^{q_T^\tau}$ defined in Lemma 5,*

$$\mathrm{TV} \left( R_K^{q_T^\tau}, Q_{t_K}^{q_T^\tau} \right)^2 \lesssim \frac{R^4 T^5 d}{\sigma_\delta^8} \bar{\gamma}_K + \frac{R^6 T^5}{\sigma_\delta^8} \bar{\gamma}_K^2 + \epsilon_{score}^2 T^3 \,..$$

*(2) If $\tau = T$ and $\beta_t = t$, we have*

$$\mathrm{TV} \left( R_K^{q_T^\tau}, Q_{t_K}^{q_T^\tau} \right)^2 \lesssim \frac{R^4 T^3 d}{\sigma_\delta^8} \bar{\gamma}_K + \frac{R^6 T^3}{\sigma_\delta^8} \bar{\gamma}_K^2 + \epsilon_{score}^2 T^2.$$

**Proof.** First, we control the discretization error in an interval $t \in [t_k, t_{k+1}]$:

$$\mathbb{E}_{Q_{t_K}^{q_T^\tau}} \left[ \left\| \mathbf{s} \left( T - t_k, \mathbf{Y}_{t_k} \right) - \nabla \ln q_{T-t} \left( \mathbf{Y}_t \right) \right\|^2 \right]$$

$$\lesssim \epsilon_{\mathrm{score}}^2 + \mathbb{E}_{Q_{t_K}^{q_T^\tau}} \left[ \left\| \nabla \ln q_{T-t_k} \left( \mathbf{Y}_{t_k} \right) - \nabla \ln q_{T-t} \left( \mathbf{Y}_{t_k} \right) \right\|^2 \right]$$

$$\qquad + \mathbb{E}_{Q_{t_K}^{q_T^\tau}} \left[ \left\| \nabla \ln q_{T-t} \left( \mathbf{Y}_{t_k} \right) - \nabla \ln q_{T-t} \left( \mathbf{Y}_t \right) \right\|^2 \right]$$

$$\lesssim \mathbb{E}_{Q_{t_K}^{q_T^\tau}} \left[ \left\| \nabla \ln \frac{q_{T-t_k}}{q_{T-t}} \left( \mathbf{Y}_{t_k} \right) \right\|^2 \right] + L^2 \mathbb{E}_{Q_{t_K}^{q_T^\tau}} \left[ \left\| \mathbf{Y}_{t_k} - \mathbf{Y}_t \right\|^2 \right] + \epsilon_{\mathrm{score}}^2$$

$$\lesssim (\tau + \beta_T) L^2 d \bar{\gamma}_K + \tau L^2 \bar{\gamma}_K^2 \left( d\tau + R^2 \right) + \tau L^3 \bar{\gamma}_K^2 + L^2 (\beta_T d \bar{\gamma}_K + R^2 \bar{\gamma}_K^2) + \epsilon_{\mathrm{score}}^2$$

$$\lesssim (\tau + \beta_T) L^2 d \bar{\gamma}_K + \tau L^2 R^2 \bar{\gamma}_K^2 + \epsilon_{\mathrm{score}}^2,$$

where $L = \max_{t \in [0, T-\delta]} \left\| \nabla^2 \log q_{T-t}(\mathbf{Y}_t) \right\| \leq \left(1 + R^2\right)/\sigma_\delta^4$ and the third inequality follows Lemma 17. Then, we know that for $\tau = T^2$ and $\beta_t = t^2$

$$\sum_{k=0}^{K-1} \mathbb{E}_{Q_{t_K}^{q_T^\tau}} \int_{t_k}^{t_{k+1}} 2\beta_{T-t} \left\| s\left(T-t_k, \mathbf{Y}_{t_k}\right) - \nabla \ln q_{T-t}(\mathbf{Y}_t) \right\|^2 \, dt$$

$$\lesssim T^5 L^2 d\bar{\gamma}_K + L^2 R^2 T^5 \bar{\gamma}_K^2 + \epsilon_{\text{score}}^2 T^3$$

$$\lesssim \frac{R^4 T^5 d}{\sigma_\delta^8} \bar{\gamma}_K + \frac{R^6 T^5}{\sigma_\delta^8} \bar{\gamma}_K^2 + \epsilon_{\text{score}}^2 T^3 \,.$$

For $\tau = T$ and $\beta_t = t$, we know that

$$\sum_{k=0}^{K-1} \mathbb{E}_{Q_{t_K}^{q_T^\tau}} \int_{t_k}^{t_{k+1}} 2\beta_{T-t} \left\| s\left(T-t_k, \mathbf{Y}_{t_k}\right) - \nabla \ln q_{T-t}(\mathbf{Y}_t) \right\|^2 \, dt$$

$$\lesssim T^3 L^2 d\bar{\gamma}_K + L^2 R^2 T^3 \bar{\gamma}_K^2 + \epsilon_{\text{score}}^2 T^2$$

$$\lesssim \frac{R^4 T^3 d}{\sigma_\delta^8} \bar{\gamma}_K + \frac{R^6 T^3}{\sigma_\delta^8} \bar{\gamma}_K^2 + \epsilon_{\text{score}}^2 T^2 \,.$$

■

Combined with the reversing beginning error controlled by Theorem 1, we can obtain the convergence guarantee for VESDE with reverse SDE.

**Corollary 1.** *Assume **Assumption 1**, **2**, **3**. Let $\gamma_K = \delta$, $\bar{\gamma}_K = argmax_{k \in \{0, \ldots, K-1\}} \gamma_k$, $\tau = T^2$ and $\beta_t = t^2$. Then, $TV\left(R_K^{q_\infty^\tau}, q_0\right)$ is bounded by*

$$\text{TV}\left(R_K^{q_\infty^\tau}, q_0\right) \leq \frac{\bar{D} \exp(-T/2)}{T} + \frac{R^2 \sqrt{d}}{\delta^6} \sqrt{\bar{\gamma}_K T^5} + \epsilon_{score} \sqrt{T^3} \,,$$

*where $\bar{D} = d|c| + \mathbb{E}[q_0] + R$. Furthermore, by choosing $\delta \leq \frac{\epsilon_{W_2}^{2/3}}{(d + R\sqrt{d})^{1/3}}$, $T \geq 2\ln \frac{\bar{D}}{\epsilon_{TV}}$, maximum stepsize $\bar{\gamma}_K \leq \delta^{12} \epsilon_{TV}^2 \ln^5(\bar{D}/\epsilon_{TV})/R^4 d$ and assuming $\epsilon_{score} \leq \widetilde{O}(\epsilon_{TV})$, the output of $R_K^{q_\infty^\tau}$ is $(\epsilon_{TV} + \epsilon_{score})$ close to $q_\delta$, which is $\epsilon_{W_2}$ close to $q_0$, with sample complexity (hiding logarithmic factors) is*

$$K \leq \widetilde{O}\left(\frac{dR^4(d + R\sqrt{d})^4}{\epsilon_{W_2}^8 \epsilon_{TV}^2}\right) \,.$$

*For choice $\beta_t = t$ and $\tau = T$, by choosing $\delta \leq \frac{\epsilon_{W_2}}{(d + R\sqrt{d})^{1/2}}$ and $\bar{\gamma}_K \leq \frac{\delta^8 \epsilon_{TV}^2 \ln^3(\bar{D}/\epsilon_{TV})}{R^4 d}$, we obtain the same sample complexity.*

**Proof.** By the data processing inequality, we know that

$$\text{TV}\left(R_K^{q_\infty^\tau}, q_0\right) \leq \text{TV}\left(R_K^{q_\infty^\tau}, R_K^{q_T^\tau}\right) + \text{TV}\left(R_K^{q_T^\tau}, Q_{t_K}^{q_T^\tau}\right)$$

$$\leq \text{TV}\left(q_T^\tau, q_\infty^\tau\right) + \text{TV}\left(R_K^{q_\infty^\tau}, Q_{t_K}^{q_T^\tau}\right) \,.$$

Then we have that for $\tau = T^2$ and $\beta_t = t^2$

$$\text{TV}\left(R_K^{q_\infty^\tau}, q_0\right) \lesssim \frac{\bar{D} \exp(-T/2)}{T} + \frac{R^2 \sqrt{d}}{\sigma_\delta^4} \sqrt{\bar{\gamma}_K T^5} + \epsilon_{\text{score}} \sqrt{T^3}$$

$$\lesssim \frac{\bar{D} \exp(-T/2)}{T} + \frac{R^2 \sqrt{d}}{\delta^6} \sqrt{\bar{\gamma}_K T^5} + \epsilon_{\text{score}} \sqrt{T^3} \,,$$

where $\bar{D} = d|c| + \mathbb{E}[q_0] + R$. The last inequality by the fact that Lemma 19. We can also use similar process to obtain the guarantee for $\tau = T$ and $\beta_t = t$

$$\text{TV}\left(R_K^{q_\infty^\tau}, q_0\right) \lesssim \frac{\bar{D} \exp(-T/2)}{\sqrt{T}} + \frac{R^2 \sqrt{d}}{\delta^4} \sqrt{\bar{\gamma}_K T^3} + \epsilon_{\text{score}} \sqrt{T^2} \,.$$

■

## C  THE PROOF OF THE CONVERGENCE GUARANTEE IN THE UNIFIED FRAMEWORK

In this work, we introduce an indicator $i \in \{1, 2\}$ for $\sigma_{T-t_K}$. We use $\tau = T^2$ as an example. When $\beta_t = t^2$ is aggressive, we choose $i = 1$, $\eta = 1$ and $\sigma_{T-t_K}^{-2}(i = 1) \leq \frac{1}{\tau} + \frac{\bar{\beta}}{\delta^3}$. When $\beta_t = t$ is conservative, we choose $i = 2$, $\eta \in [0, 1)$ and $\sigma_{T-t_K}^{-2}(i = 2) \leq \frac{1}{\tau} + \frac{\bar{\beta}}{\delta^2}$. In the proof process of Lemma 2, Lemma 7, Lemma 8 and Lemma 9, we ignore the indicator $i$ since this lemma does not involve the specific value of $\sigma_{T-t_K}^2(i)$. Before the proof of this section, we first recall the stochastic flow of the reverse process for any $x \in \mathbb{R}^d$ and $s, t \in [0, T]$ with $t \geq s$:

$$\mathrm{d}\mathbf{Y}_{s,t}^x = \beta_{T-t} \left\{ \mathbf{Y}_{s,t}^x/\tau + \left(1 + \eta^2\right) \nabla \log q_{T-t} \left(\mathbf{Y}_{s,t}^x\right) \right\} \mathrm{d}t + \eta\sqrt{2\beta_{T-t}}\mathrm{d}\mathbf{B}_t, \qquad \mathbf{Y}_{s,s}^x = x,$$

and the interpolation of its discretization for any $k \in \{0, ..., K\}$ and $t \in [s_k, t_{k+1})$:

$$\mathrm{d}\bar{\mathbf{Y}}_{s,t}^x(k) = \beta_{T-t} \left\{ \bar{\mathbf{Y}}_{s,t}^x/\tau + \left(1 + \eta^2\right) \mathbf{s} \left(T - s_k, \bar{\mathbf{Y}}_{s,t}^x\right) \right\} \mathrm{d}t + \eta\sqrt{2\beta_{T-t}}\mathrm{d}\mathbf{B}_t, \qquad \bar{\mathbf{Y}}_{s,s}^x = x,$$

where $s_k = \max(s, t_k)$. To deal with the discretization error, we use the approximation technique used in Bortoli (2022). Hence, we introduce the tangent process:

$$\mathrm{d}\nabla\mathbf{Y}_{s,t}^x = \beta_{T-t} \left\{ \mathbf{I}/\tau + \left(1 + \eta^2\right) \nabla^2 \log q_{T-t}(\mathbf{Y}_{s,t}^x) \right\} \nabla\mathbf{Y}_{s,t}^x \mathrm{d}t, \qquad \nabla\mathbf{Y}_{s,s}^x = \mathbf{I}.$$

Then, we discuss the interpolation formula, which is used to control the discretization error.

**Proposition 1.** *For $s, t \in [0, T]$ with $s < t$, any $k \in \{0, ..., K\}$ and $(\omega_v)_{v \in [s,T]}$, we define that*

$$b_u(\omega) = \beta_{T-u}(\frac{1}{\tau}\omega_u + (1 + \eta^2)\nabla \log q_{T-u}(\omega_u)),$$

$$\bar{b}_u(\omega) = \beta_{T-u}(\frac{1}{\tau}\omega_u + (1 + \eta^2)\mathbf{s}(T - s_k, \omega_{s_k})), \quad \Delta b_u(\omega) = b_u(\omega) - \bar{b}_u(\omega),$$

*where $s_k = \max(s, t_k)$ and $u \in [s_k, t_{k+1}]$. Then, for any $x \in \mathbb{R}^d$, we have that*

$$\mathbf{Y}_{s,t}^x - \bar{\mathbf{Y}}_{s,t}^x = \int_s^t \nabla\mathbf{Y}_{u,t}^x \left(\bar{\mathbf{Y}}_{s,u}^x\right)^\top \Delta b_u \left(\left(\bar{\mathbf{Y}}_{s,v}^x\right)_{v \in [s,T]}\right) \mathrm{d}u,$$

*where for any $u \in [0, T)$, there exists a $k \in \{0, ..., K\}$ satisfies $u \in [s_k, t_{k+1})$.*

For reverse SDE, the augmentation is similar to Bortoli (2022) (Appendix E). When $\eta = 0$, the stochastic extension of the Alekseev–Gröbner formula (Del Moral and Singh, 2022) degenerates into the original version (Alekseev, 1961). After that, we control the tangent process.

**Lemma 2.** *Assume **Assumption 1** and **2**. Then, for $\forall s \in [0, t_K], x \in \mathbb{R}^d$, and $i \in \{1, 2\}$, we have*

$$\|\nabla\mathbf{Y}_{s,t_K,i}^x\| \leq \exp\left[\frac{R^2}{2\sigma_{T-t_K}^2(i)} + \frac{(1 - \eta^2)}{2} \int_0^{t_K} \frac{\beta_{T-u}}{\tau}\mathrm{d}u\right].$$

*Furthermore, if assuming $\left\|\nabla^2 \log q_t\left(x_t\right)\right\| \leq \Gamma/\sigma_t^2$, we have that*

$$\|\nabla\mathbf{Y}_{s,t_K,i}^x\| \leq \sigma_{T-t_K}^{-(1+\eta^2)\Gamma}(i) \exp\left[\left(\left(1 + \eta^2\right)\Gamma + 2\right) \int_0^{t_K} \frac{\beta_{T-u}}{\tau}\mathrm{d}u\right].$$

**Proof.** Using Eq. (7) and Lemma 13, we have

$$\mathrm{d}\left\|\nabla\mathbf{Y}_{s,t}^x\right\|^2$$

$$\leq 2\beta_{T-t}\left(\frac{1}{\tau}\left\|\nabla\mathbf{Y}_{s,t}^x\right\|^2 - \left(1 + \eta^2\right)\left(1 - m_{T-t}^2 R^2/\left(2\sigma_{T-t}^2\right)\right)/\sigma_{T-t}^2\left\|\nabla\mathbf{Y}_{s,t}^x\right\|^2\right)\mathrm{d}t.$$

Using Lemma 18, we have

$$\int_s^t \beta_{T-u}\left(\frac{1}{\tau} - \left(1 + \eta^2\right)/\sigma_{T-u}^2 + \left(1 + \eta^2\right) m_{T-u}^2 R^2/2\sigma_{T-u}^4\right)\mathrm{d}u$$

$$\leq \left(\left(1 + \eta^2\right) R^2/4\right)\left(\sigma_{T-t}^{-2} - \sigma_{T-s}^{-2}\right) + \frac{1 - \eta^2}{2}\int_s^t \frac{\beta_{T-u}}{\tau}\mathrm{d}u$$

$$\leq \frac{\left(1 + \eta^2\right) R^2}{4\sigma_{T-t}^2} + \frac{1 - \eta^2}{2}\int_s^t \frac{\beta_{T-u}}{\tau}\mathrm{d}u.$$

Note that $\nabla \mathbf{Y}_{s,s} = \mathbf{I}$, we get

$$\|\nabla \mathbf{Y}_{s,t_K}^x\|^2 \leq \exp\left[\frac{\left(1+\eta^2\right)R^2}{2\sigma_{T-t}^2} + \left(1-\eta^2\right)\int_0^{t_K}\frac{\beta_{T-u}}{\tau}\mathrm{d}u\right].$$

When we assume $\left\|\nabla \log q_t^2\left(x_t\right)\right\| \leq \Gamma/\sigma_t^2$, we know that

$$\mathrm{d}\left\|\nabla \mathbf{Y}_{s,t}^x\right\|^2 \leq 2\beta_{T-t}\left(\frac{1}{\tau} - \frac{\left(1+\eta^2\right)\Gamma}{\sigma_{T-t}^2}\right)\left\|\nabla \mathbf{Y}_{s,t}^x\right\|^2 \mathrm{d}t.$$

Using Lemma 18, we have

$$2\int_s^t \beta_{T-u}/\sigma_{T-u}^2 \mathrm{d}u$$
$$\leq \log\left(\exp\left[2\int_0^{T-s}\frac{\beta_{T-u}}{\tau}\,\mathrm{d}u\right]-1\right) - \log\left(\exp\left[2\int_0^{T-t}\frac{\beta_{T-u}}{\tau}\,\mathrm{d}u\right]-1\right)$$
$$\leq \log\left(\sigma_{T-s}^2\right) - \log\left(\sigma_{T-t}^2\right) + \int_{T-t}^{T-s}\frac{\beta_u}{\tau}\,\mathrm{d}u.$$

Then we have

$$\|\nabla \mathbf{Y}_{s,t_K}^x\|^2 \leq \sigma_{T-t_K}^{-(1+\eta^2)\Gamma}\exp\left[\left(\left(1+\eta^2\right)\Gamma+2\right)\int_0^{t_K}\frac{\beta_{T-u}}{\tau}\mathrm{d}u\right].$$

Thus we complete our proof. ∎

After bounding the gradient of the tangent process, the remaining term is $\|\Delta b\|$:

$$\|\Delta b\| \leq \|\Delta^{(a,b)}b\| + \|\Delta^{(b,c)}b\| + \|\Delta^{(c,d)}b\|, \tag{8}$$

where $b^{(a)} = b$ and $b^{(d)} = \bar{b}$. Moreover,

$$b_u^{(b)}(\omega) = \beta_{T-u}(\frac{1}{\tau}\omega_u + (1+\eta^2)\nabla\log q_{T-s_k}(\omega_u)),$$
$$b_u^{(c)}(\omega) = \beta_{T-u}(\frac{1}{\tau}\omega_u + (1+\eta^2)\nabla\log q_{T-s_k}(\omega_{s_k})),$$
$$\Delta_b^{a,b} = b^{(a)} - b^{(b)}, \; \Delta_b^{b,c} = b^{(b)} - b^{(c)}, \; \Delta_b^{c,d} = b^{(c)} - b^{(d)}.$$

We then control $\|\Delta^{(a,b)}b\|, \|\Delta^{(b,c)}b\|, \|\Delta^{(c,d)}b\|$ separately. In this section, $\|\Delta^{(c,d)}b\| = 0$ since we assume that the accurate score function is achieved. For $\left\|\Delta^{(a,b)}b_u(\omega)\right\|$, we have the following lemma.

**Lemma 7.** *For $s, u \in [0, T)$ such that $u \geq s, u \in [s_k, t_{k+1})$ and $\omega = (\omega_v)_{v\in[s,T]}$ we have*

$$\|\Delta^{(a,b)}b_u\left(\omega\right)\|$$
$$\leq \left(1+\eta^2\right)\beta_{T-u}\sup_{v\in[T-u,T-t_k]}\left(\beta_v/\sigma_v^6\right)\left(2+R^2\right)\left(R+\|\omega_u\|\right)\gamma_k.$$

**Proof.** Without loss of generality, we assume $s \leq t_k$. Then

$$\|\Delta^{(a,b)}b_u\left(\omega\right)\| \leq \left(1+\eta^2\right)\beta_{T-u}\|\nabla\log q_{T-u}\left(\omega_u\right) - \nabla\log q_{T-t_k}\left(\omega_u\right)\|$$
$$\leq \left(1+\eta^2\right)\beta_{T-u}\gamma_k\sup_{v\in[T-u,T-t_k]}\|\partial_v\nabla\log q_{T-v}\left(\omega_u\right)\|.$$

Then by Lemma 16, we have

$$\|\Delta^{(a,b)}b_u\left(\omega\right)\|$$
$$\leq \left(1+\eta^2\right)\beta_{T-u}\sup_{v\in[T-u,T-t_k]}\left(\beta_v/\sigma_v^6\right)\left(2+R^2\right)\left(R+\|\omega_u\|\right)\gamma_k.$$

∎

For $\left\|\Delta^{(b,c)}b_u(\omega)\right\|$, we have the following lemma.

**Lemma 8.** *For $s, u \in [0, T)$ such that $u \geq s, u \in [s_k, t_{k+1})$ and $\omega = (\omega_v)_{v \in [s,T]}$ we have*

$$\|\Delta^{(b,c)}b_u(\omega)\| \leq \left(1+\eta^2\right)\left(\beta_{T-u}/\sigma_{T-u}^4\right)\left(1+R^2\right)\|\omega_u - \omega_{s_k}\|.$$

**Proof.** Without loss of generality, we assume $s \leq t_k$. In this case $s_k = t_k$, Then

$$\|\Delta^{(b,c)}b_u(\omega)\| \leq \left(1+\eta^2\right)\beta_{T-u}\|\nabla\log q_{T-t_k}(\omega_{t_k}) - \nabla\log q_{T-t_k}(\omega_u)\|$$
$$\leq \left(1+\eta^2\right)\beta_{T-u}\sup_{v\in[u,T-t_k]}\|\nabla^2\log q_{T-t_k}(\omega_v)\|\|\omega_u - \omega_{t_k}\|.$$

Using Lemma 14, we have that

$$\|\Delta^{(b,c)}b_u(\omega)\| \leq \left(1+\eta^2\right)\left(\beta_{T-u}/\sigma_{T-u}^4\right)\left(1+R^2\right)\|\omega_u - \omega_{t_k}\|.$$

Then the proof is complete. ∎

We need to control the reverse process when dealing with $\Delta b$. The following lemma shows an upper bound for the reverse $Y_k$.

**Lemma 9.** *Assume **Assumption 1**, **Assumption 2**, and there exists $\delta > 0$ such that $\frac{\gamma_k\beta_{T-t_k}}{\sigma_{T-t_k}^2} \leq \delta \leq$
$1/28$ for any $k \in \{0, \cdots, K\}$, then we have*

$$\mathbb{E}[\|Y_k\|^2] \leq U(\tau) = \tau d + B(1/A + \delta),$$

*where*

$$A = 4\eta^2 + 2 - 2\delta - 4(1+\eta^2)(1+\delta)\mu R$$

$$B = 4(1+\eta^2)R^2\delta + 2(1+\eta^2)(1+\delta)\frac{R}{\mu} + 4\eta^2\tau d$$

*and $\mu$ is an arbitrary positive number which makes $A > 0$. In particular, if $\delta \leq 1/28$, then*

$$\mathbb{E}[\|Y_k\|^2] \leq U_0(\tau) = 111R^2 + 13\tau d.$$

**Proof.** Recall the discretization of the backward process (the explicit form of Eq. (6))

$$Y_{k+1} = Y_k + \gamma_{1,k}\left(\frac{1}{\tau}Y_k + (1+\eta^2)\mathbf{s}(T - t_k, Y_k)\right) + \eta\sqrt{2\gamma_{2,k}}Z_k,$$

$$\gamma_{1,k} = \exp\left[\int_{T-t_{k+1}}^{T-t_k}\beta_s\,\mathrm{d}s\right] - 1, \quad \gamma_{2,k} = \left(\exp\left[2\int_{T-t_{k+1}}^{T-t_k}\beta_s\,\mathrm{d}s\right] - 1\right)/2,$$

where $\{Z_k\}_{k \in K}$ are independent Gaussian random variables. It is clear that $\gamma_{1,k} \leq \gamma_{2,k} \leq 2\gamma_{1,k}$, and using Lemma 13 we have

$$\langle x_t, \mathbf{s}(t, x_t)\rangle = \langle x_t, \nabla\log q_t(x_t)\rangle$$
$$\leq -\|x_t\|^2/\sigma_t^2 + m_t R\|x_t\|/\sigma_t^2$$
$$\leq (-1 + \mu m_t R)\|x_t\|^2/\sigma_t^2 + (m_t R/\mu)/\sigma_t^2,$$

where the first equality follows that we assume the accurate score function. For any $\mu > 0$. Again using Lemma 13, we have

$$\|\mathbf{s}(t, x_t)\|^2 = \|\nabla\log q_t(x_t)\|^2$$
$$\leq 2\|x_t\|^2/\sigma_t^4 + 2m_t^2 R^2/\sigma_t^4.$$

Combining the results above, we have

$$\mathbb{E}[\|Y_{k+1}\|^2] = (1 + \frac{\gamma_{1,k}}{\tau})^2\mathbb{E}[\|Y_k\|^2] + (1+\eta^2)^2\gamma_{1,k}^2\mathbb{E}[\|s(T - t_k, Y_k)\|^2]$$

$$+ 2(1+\eta^2)(1 + \frac{\gamma_{1,k}}{\tau})\gamma_{1,k}\mathbb{E}[\langle Y_k, s(T - t_k, Y_k)\rangle] + 2\eta^2\gamma_{2,k}d$$

$$\leq ((1 + \frac{\gamma_{1,k}}{\tau})^2 + 2(1+\eta^2)^2\gamma_{1,k}^2/\sigma_{T-t_k}^4$$

$$+ 2(1+\eta^2)(1 + \frac{\gamma_{1,k}}{\tau})\gamma_{1,k}(-1 + \mu m_{T-t_k}R)/\sigma_{T-t_k}^2)\mathbb{E}[\|Y_k\|^2]$$

$$+ \frac{2m_{T-t_k}^2 R^2}{\sigma_{T-t_k}^4}(1+\eta^2)^2\gamma_{1,k}^2 + \frac{m_{T-t_k}R}{\mu\sigma_{T-t_k}^2}(1+\eta^2)(1 + \frac{\gamma_{1,k}}{\tau})\gamma_{1,k} + 4\eta^2\gamma_{1,k}d.$$

If we denote $\delta_k = \gamma_{1,k}/\sigma_{T-t_k}^2$ and notice the fact that $m_t \in [0,1], \sigma_t^2 \in [0,\tau], \eta \in [0,1]$, then we have

$$\mathbb{E}[\|Y_{k+1}\|^2] \leq (1 + 2\delta_k + \delta_k^2)\mathbb{E}[\|Y_k\|^2] + 8\delta_k^2\mathbb{E}[\|Y_k\|^2]$$
$$+ 2(1+\delta_k)\delta_k(-1+\mu R)\mathbb{E}[\|Y_k\|^2] + 8R^2\delta_k^2 + \frac{2R}{\mu}\delta_k(1+\delta_k) + 4\tau\delta_k d.$$

We also have that

$$\gamma_{1,k} = \exp[\int_{T-t_{k+1}}^{T-t_k} \beta_s \mathrm{d}s] - 1 \leq \exp[\beta_{T-t_k}\gamma_k] - 1 \leq 2\beta_{T-t_k}\gamma_k,$$

where the last inequality follows that $\gamma_k = \exp(-T)$, $\beta_{T-t_k}\gamma_k \leq 1/2$ for small enough stepsize, and $e^\omega - 1 \leq 2\omega$ for any $\omega \in [0, 1/2]$. We get $\delta_k \leq 2\gamma_k\beta_{T-t_k}/\sigma_{T-t_k}^2 \leq 2\delta$. Thus

$$\mathbb{E}[\|Y_{k+1}\|^2] \leq (1 + 2\delta_k + 2\delta_k\delta)\mathbb{E}[\|Y_k\|^2] + 16\delta_k\delta\mathbb{E}[\|Y_k\|^2]$$
$$+ 4(1+\delta)(-1+\mu R)\delta_k\mathbb{E}[\|Y_k\|^2] + 16R^2\delta_k\delta + 4(1+\delta)\frac{R}{\mu}\delta_k + 4\tau d\delta_k.$$

Hence, we have

$$\mathbb{E}[\|Y_{k+1}\|^2] \leq (1 + \delta_k[-2 + 14\delta + 4(1+\delta)\mu R])\mathbb{E}[\|Y_k\|^2]$$
$$+ \delta_k[16R^2\delta + 4(1+\delta)\frac{R}{\mu} + 4\tau d].$$

We denote $A = 2 - 14\delta - 4(1+\delta)\mu R$ and $B = 16R^2\delta + 4(1+\delta)\frac{R}{\mu} + 4\tau d$, then

$$\mathbb{E}[\|Y_{k+1}\|^2] \leq (1 - \delta_k A)\mathbb{E}[\|Y_k\|^2] + \delta_k B.$$

Notice that $\mathbb{E}[\|Y_0\|^2] = d\tau$ and if $\mathbb{E}[\|Y_k\|^2] \geq B/A$ it is decreasing, if $\mathbb{E}[\|Y_k\|^2] \leq B/A$ we have $\mathbb{E}[\|Y_{k+1}\|^2] \leq B/A + \delta B$. so

$$\mathbb{E}[\|Y_k\|^2] \leq \tau d + B(1/A + \delta).$$

Notice that when $\delta \leq 1/28$, if we choose $\mu = 1/(4(1+\delta)R)$, $A \geq 1/2$, and

$$B \leq 37R^2 + 4\tau d.$$

Then, the proof is complete. $\blacksquare$

The following lemma shows a discretization error in the $k$-the interval.

**Lemma 10.** *Assume* **Assumption 1**, **Assumption 2** *and* $\gamma_k\beta_{T-t_k}/\sigma_{T-t_k}^2 \leq 1/28$ *for any* $k \in \{0, \cdots, K-1\}$. *Then for any* $k$, $t \in [t_k, t_{k+1}]$ *and* $i \in \{1, 2\}$, *we have that*

$$\mathbb{E}[\|\bar{\mathbf{Y}}_t - \bar{\mathbf{Y}}_{t_k}\|^2] \leq L_i(\tau)\beta_{T-t_k}\gamma_k,$$

*where* $L_i(\tau) = \bar{\gamma}_K\kappa_i(\tau)(\frac{64}{\sigma_{T-t_K}^2(i)} + \frac{8}{\tau})U_0(\tau) + 64R^2\frac{\bar{\gamma}_K\kappa_i(\tau)}{\sigma_{T-t_K}^2(i)} + 4d$, $\bar{\gamma}_K$, $\kappa_i(\tau)$ *is defined in Lemma 11 and* $U_0(\tau)$ *is defined in Lemma 9.*

**Proof.** Recall the discretized backward process

$$\bar{\mathbf{Y}}_t = \bar{\mathbf{Y}}_{t_k} + (\exp[\int_{T-t}^{T-t_k} \beta_s \mathrm{d}s] - 1)(\frac{1}{\tau}\bar{\mathbf{Y}}_{t_k} + (1+\eta^2)\mathbf{s}(T-t_k, \bar{\mathbf{Y}}_{t_k}))$$
$$+ \eta(\exp[2\int_{T-t}^{T-t_k} \beta_s \mathrm{d}s - 1])^{1/2}Z,$$

where $Z$ is a standard Gaussian random variable. By directly calculating, we have that

$$\mathbb{E}[\|\bar{\mathbf{Y}}_t - \bar{\mathbf{Y}}_{t_k}\|^2] = 2(\exp[\int_{T-t}^{T-t_k} \beta_s \mathrm{d}s] - 1)^2(\frac{1}{\tau^2}\mathbb{E}[\|\bar{\mathbf{Y}}_{t_k}\|^2] + (1+\eta^2)^2\mathbb{E}[\|s(T-t_k, \bar{\mathbf{Y}}_{t_k})\|^2])$$
$$+ \eta^2(\exp[2\int_{T-t}^{T-t_k} \beta_s \mathrm{d}s] - 1)d.$$

By Lemma 13 and accurate score function assumption,

$$\|\mathbf{s}(T - t_k, \bar{\mathbf{Y}}_{t_k})\|^2 \leq 2\|\bar{\mathbf{Y}}_{t_k}\|^2/\sigma^4_{T-t_k}(i) + 2m^2_{T-t_k}R^2/\sigma^4_{T-t_k}(i).$$

So we have that

$$\mathbb{E}[\|\bar{\mathbf{Y}}_t - \bar{\mathbf{Y}}_{t_k}\|^2] \leq 2(\exp[\int_{T-t}^{T-t_k}\beta_s \mathrm{d}s] - 1)^2((\frac{8}{\sigma^4_{T-t_k}(i)} + \frac{1}{\tau^2})\mathbb{E}[\|\bar{\mathbf{Y}}_{t_k}\|^2] + \frac{8R^2}{\sigma^4_{T-t_k}(i)})$$
$$+ (\exp[2\int_{T-t}^{T-t_k}\beta_s \mathrm{d}s] - 1)d.$$

By $e^{2w} - 1 \leq 1 + 4w$ for any $w \in [0, 1/2]$ and $\gamma_k \sup_{v \in [T-t_{k+1}, T-t_k]}\beta_v/\sigma^2_v \leq 1/28$ for any $k \in \{0, ..., K-1\}$, we have

$$\exp[\rho\int_{T-t}^{T-t_k}\beta_s \mathrm{d}s] - 1 \leq 2\rho\beta_{T-t_k}\gamma_k.$$

for $\rho = 1, 2$. And using Lemma 9 and Lemma 19 we have

$$\mathbb{E}[\|\bar{\mathbf{Y}}_t - \bar{\mathbf{Y}}_{t_k}\|^2]$$
$$\leq (\frac{64\gamma_k}{\sigma^4_{T-t_k}(i)} + \frac{8\beta_{T-t_k}\gamma_k}{\tau^2})U_0(\tau)\beta_{T-t_k}\gamma_k + 64R^2\frac{\gamma_k}{\sigma^4_{T-t_k}(i)}\beta_{T-t_k}\gamma_k + 4d\beta_{T-t_k}\gamma_k.$$

We denote $L_i(\tau) = \bar{\gamma}_K\kappa_i(\tau)(\frac{64}{\sigma^2_{T-t_K}(i)} + \frac{8}{\tau})U_0(\tau) + 64R^2\frac{\bar{\gamma}_K\kappa_i(\tau)}{\sigma^2_{T-t_K}(i)} + 4d$ for $i \in \{1, 2\}$ and the proof is complete. ∎

**Lemma 11.** *Assume* **Assumption 1** *and* **Assumption 2**, $\gamma_k \sup_{v \in [T-t_{k+1}, T-t_k]}\beta_v/\sigma^2_v \leq 1/28$ *for any* $k \in \{0, ..., K-1\}$. *Let* $\bar{\gamma}_K = argmax_{k \in \{0,...,K-1\}}\gamma_k$, $\kappa_i(\tau) = \max\{\bar{\beta}, \frac{T^2}{T^{-1+i}}\}\sigma^{-2}_{T-t_K}(i)$, *and*

$$C_i(\tau) = 2(2 + R^2)(R + U_0^{1/2}(\tau)) + 2L_i^{1/2}(\tau)\tau^{3/2}(1 + R^2),$$

*for* $i \in \{1, 2\}$. *Then, for any* $s, u \in [0, t_K]$ *with* $u \geq s$ *and* $i \in \{1, 2\}$, *we have*

$$\mathbb{E}[\|\Delta b_{u,i}((\bar{\mathbf{Y}}_{s,v})_{v \in [s,T]})\|] \leq C_i(\tau)[\kappa_i^2(\tau)\sigma^{-2}_{T-t_K}(i)\bar{\gamma}_K^{1/2} + \kappa_i^2(\tau)]\bar{\gamma}_K^{1/2},$$

*where* $\bar{\mathbf{Y}}_{s,s} \sim \mathrm{N}(0, \mathbf{I})$.

**Proof.** Combining Lemma 7, Lemma 8 and the exact score function, we get

$$\|\Delta b_{u,i}(\omega)\| \leq (1 + \eta^2)\sup_{v \in [T-t_{k+1}, T-t_k]}(\beta_v^2/\sigma_v^6(i))(2 + R^2)(\mathrm{diam}(\mathcal{M} + \|\omega_u\|))\gamma_k$$
$$+ (1 + \eta^2)(\beta_{T-u}/\sigma^4_{T-u}(i))(1 + \mathrm{diam}(\mathcal{M}^2))\|\omega_u - \omega_{s_k}\|.$$

For any $u \in [T - t_K, T]$, using Lemma 20 we have $\beta_u/\sigma^2_u(i) \leq \kappa_i(\tau)$. Hence,

$$\|\Delta b_{u,i}(\omega)\| \leq (1 + \eta^2)\sup_{v \in [T-t_{k+1}, T-t_k]}(\beta_v^2/\sigma_v^6(i))(2 + \mathrm{diam}(\mathcal{M}^2))(R + \|\omega_u\|)\gamma_k$$
$$+ (1 + \eta^2)(\beta_{T-u}/\sigma^4_{T-u}(i))(1 + \mathrm{diam}(\mathcal{M}^2))(\|\omega_u - \omega_{t_k}\|)$$
$$\leq (1 + \eta^2)(\kappa_i^2(\tau)/\sigma^2_{T-t_{k+1}}(i))\gamma_k(2 + \mathrm{diam}(\mathcal{M}^2))(R + \|\omega_u\|)$$
$$+ (1 + \eta^2)\kappa_i^2(\tau)(1 + R^2)\|\omega_u - \omega_{t_k}\|/\beta_{T-u}.$$

Combining this with Lemma 9 and Lemma 10,

$$\mathbb{E}[\|\Delta b_{u,i}((\bar{\mathbf{Y}}_{s,v})_{v \in [s,T]})\|] \leq (1 + \eta^2)(\kappa_i^2(\tau)/\sigma^2_{T-t_{k+1}}(i))\bar{\gamma}_K(2 + R^2)(R + U_0^{1/2}(\tau))$$
$$+ (1 + \eta^2)\kappa_i^2(\tau)(1 + R^2)L_i^{1/2}(\tau)\max\{\bar{\beta}, \tau\}^{3/2}\bar{\gamma}_K^{1/2}.$$

We denote $C_i(\tau) = 2(2 + R^2)(R + U_0^{1/2}(\tau)) + 2L_i^{1/2}(\tau)\tau^{3/2}(1 + R^2)$, for $i \in \{1, 2\}$, then we have

$$\mathbb{E}[\|\Delta b_{u,i}((\bar{\mathbf{Y}}_{s,v})_{v \in [s,T]})\|] \leq C_i(\tau)((\kappa_i^2(\tau)/\sigma^2_{T-t_{k+1}})\bar{\gamma}_K + \kappa_i^2(\tau)\bar{\gamma}_K^{1/2}).$$

∎

**Lemma 12.** *Assume **Assumption 1** and **Assumption 2**, $\gamma_k \sup_{v \in [T-t_{k+1}, T-t_k]} \beta_v / \sigma_v^2 \leq 1/28$ for any $k \in \{0, ..., K-1\}$. Let $\bar{\gamma}_K = argmax_{k \in \{0,...,K-1\}} \gamma_k$, $\gamma_K = \delta$, and $\delta \leq 1/32$. Then*

$$W_1\left(R_K^{q_\infty^\tau}, Q_{t_K}^{q_\infty^\tau}\right) \leq C_i(\tau)\kappa_i^2(\tau)T \exp\left[\frac{R^2}{2\sigma_{T-t_K}^2(i)} + \frac{(1-\eta^2)}{2}\right][\frac{\bar{\gamma}_K^{1/2}}{\sigma_{T-t_K}^2(i)} + 1]\bar{\gamma}_K^{1/2},$$

*where $C_i(\tau), \kappa_i(\tau)$ for $i \in \{1, 2\}$ are the same terms to Theorem 2.*

**Proof.** By **Proposition 1** we have

$$\|\mathbf{Y}_{t_K} - Y_K\| = \|\mathbf{Y}_{t_K} - \bar{\mathbf{Y}}_{t_K}\| \leq \int_0^{t_K} \|\nabla\mathbf{Y}_{u,t_K,i}(\bar{\mathbf{Y}}_{0,u})\|\|\Delta b_{u,i}((\bar{\mathbf{Y}}_{0,v})_{v \in [0,T]})\| \mathrm{d}u.$$

$$\|\mathbf{Y}_{t_K} - Y_K\|$$
$$\leq \exp\left[\frac{(1+\eta^2)R^2}{4\sigma_{T-t}^2(i)} + \frac{(1-\eta^2)}{2}\int_0^{t_K}\frac{\beta_{T-u}}{\tau}\mathrm{d}u\right]\int_0^{t_K}\|\Delta b_{u,i}((\bar{\mathbf{Y}}_{0,v})_{v \in [0,T]})\|\mathrm{d}u.$$

Then by definition of Wasserstein distance, we have

$$W_1(q_\infty Q_{t_K}, q_\infty R_K)$$
$$\leq \mathbb{E}[\|\mathbf{Y}_{t_K} - Y_K\|]$$
$$\leq \exp\left[\frac{(1+\eta^2)R^2}{4\sigma_{T-t_K}^2(i)} + \frac{(1-\eta^2)}{2}\int_0^{t_K}\frac{\beta_{T-u}}{\tau}\mathrm{d}u\right]\int_0^{t_K}\mathbb{E}[\|\Delta b_{u,i}((\bar{\mathbf{Y}}_{0,v})_{v \in [0,T]})\|]\mathrm{d}u$$
$$\leq C_i(\tau)T \exp\left[\frac{(1+\eta^2)R^2}{4\sigma_{T-t_K}^2(i)} + \frac{(1-\eta^2)}{2}\right][\kappa_i^2(\tau)\sigma_{T-t_K}^{-2}(i)\bar{\gamma}_K^{1/2} + \kappa_i^2(\tau)]\bar{\gamma}_K^{1/2}.$$

$\blacksquare$

**Theorem 2.** *Assume **Assumption 1** and 2, $\gamma_k \sup_{v \in [T-t_{k+1}, T-t_k]} \beta_v / \sigma_v^2 \leq 1/28$ for $\forall k \in \{0, ..., K-1\}$, and $\delta \leq 1/32$. Let $\bar{\gamma}_K = argmax_{k \in \{0,...,K-1\}} \gamma_k$ and $\gamma_K = \delta$. Then, for $\forall \tau \in [T, T^2]$, we have the following convergence guarantee.*

*(1) If $\eta = 1$ (the reverse SDE), choosing an aggressive $\beta_t = t^2$, we have*

$$W_1\left(R_K^{q_\infty^\tau}, q_0\right) \leq C_1(\tau)T \exp\left[\frac{R^2}{2}(\frac{\bar{\beta}}{\delta^3} + \frac{1}{\tau})\right][\kappa_1^2(\tau)(\frac{\bar{\beta}}{\delta^3} + \frac{1}{\tau})\bar{\gamma}_K^{1/2} + \kappa_1^2(\tau)]\bar{\gamma}_K^{1/2}$$
$$+ \exp\left[\frac{R^2}{2}(\frac{\bar{\beta}}{\delta^3} + \frac{1}{\tau})\right]\frac{\bar{D}\exp(-T/2)}{\sqrt{\tau}} + 2(\frac{R}{\tau} + \sqrt{d})\sqrt{\delta},$$

*where $C_1(\tau)$ is linear in $\tau^2$, $\kappa_1(\tau) = \max\{\bar{\beta}, T^2\}(1/\tau + \bar{\beta}/\delta^3)$.*

*(2) If $\eta = 0$ (the reverse PFODE), choosing a conservative $\beta_t$ satisfies **Assumption 1**, we have*

$$W_1\left(R_K^{q_\infty^\tau}, q_0\right) \leq C_2(\tau)T \exp\left[\frac{R^2}{2}(\frac{\bar{\beta}}{\delta^2} + \frac{1}{\tau}) + \frac{1}{2}\right][\kappa_2^2(\tau)(\frac{\bar{\beta}}{\delta^2} + \frac{1}{\tau})\bar{\gamma}_K^{1/2} + \kappa_2^2(\tau)]\bar{\gamma}_K^{1/2}$$
$$+ \exp\left[\frac{R^2}{2}(\frac{\bar{\beta}}{\delta^2} + \frac{1}{\tau})\right]\frac{\bar{D}}{\sqrt{\tau}} + 2(\frac{R}{\tau} + \sqrt{d})\sqrt{\delta},$$

*where $C_2(\tau)$ is linear in $\tau^2$, $\kappa_2(\tau) = \max\{\bar{\beta}, T\}(1/\tau + \bar{\beta}/\delta^2)$.*

**Proof.** To obtain the convergence guarantee, we need to control three error terms:

$$W_1\left(R_K^{q_\infty^\tau}, q_0\right) \leq W_1\left(R_K^{q_\infty^\tau}, Q_{t_K}^{q_\infty^\tau}\right) + W_1\left(Q_{t_K}^{q_\infty^\tau}, Q_{t_K}^{q_0 P_T}\right) + W_1\left(Q_{t_K}^{q_0 P_T}, q_0\right).$$

For term $W_1\left(R_K^{q_\infty^\tau}, Q_{t_K}^{q_\infty^\tau}\right)$, we use Lemma 12.

For the second term, we define $\left(\mathbf{Y}_{0,t}^{x}\right)_{t\in[0,T]}$ and $\left(\mathbf{Y}_{0,t}^{y}\right)_{t\in[0,T]}$ be the reverse processes with initial condition $x$ and $y$. Then we have

$$\|\mathbf{Y}_{0,t}^{x} - \mathbf{Y}_{0,t}^{y}\| \leq \|x - y\| \int_0^1 \|\nabla \mathbf{Y}_{0,t}^{z_\lambda}\| d\lambda \,,$$

where $z_\lambda = \lambda x + (1 - \lambda)y$. In this work, we choose $x \sim q_\infty^\tau$ and $y \sim q_0 P_T$. Combined with the above inequality, Theorem 1 and Lemma 2, we know that:

$$W_1\left(Q_{t_K}^{q_\infty^\tau}, Q_{t_K}^{q_0 P_T}\right)$$

$$\leq \exp\left[\frac{R^2}{2\sigma_{T-t_K}^2(i)} + \frac{(1-\eta^2)}{2}\int_0^{t_K} \frac{\beta_{T-u}}{\tau} du\right] \|q_0 P_T - q_\infty^\tau\|$$

$$\leq \frac{\sqrt{m_T}\bar{D}}{\sigma_T} \exp\left[\frac{R^2}{2\sigma_{T-t_K}^2(i)} + \frac{(1-\eta^2)}{2}\int_0^{t_K} \frac{\beta_{T-u}}{\tau} du\right] \,.$$

For the last term, we use exactly the same process with Bortoli (2022) with bounded $\sigma_{T-t_K}^2$:

$$W_1\left(Q_{t_K}^{q_0 P_T}, q_0\right) \leq \mathbb{E}\left[\|X - m_{T-t_K}X + \sigma_{T-t_K}Z\|\right]$$

$$\leq (\frac{R}{\tau} + \sqrt{d})\sigma_{T-t_K}$$

$$\leq 2(\frac{R}{\tau} + \sqrt{d})\sqrt{\delta} \,,$$

where the second inequality follows that $\sigma_{T-t_K}^2 + \tau m_{T-t_K} = \tau$. ∎

In the end of the section, we provide the proof of Corollary 3.

**Corollary 3.** *Assume **Assumption 1**, **Assumption 2** and $\|\nabla^2 \log q_t(x_t)\| \leq \Gamma/\sigma_t^2$. Let $\eta = 0$ (reverse PFODE), $\epsilon \in (0, 1/32), \tau = T^2, \beta_t = t, \bar{\gamma}_K = argmax_{k\in\{0,\ldots,K-1\}}\gamma_k, \gamma_K = \delta$, we have*

$$W_1\left(R_K^{q_\infty^\tau}, q_0\right) \leq C_2(\tau)T\frac{\bar{\beta}^{\frac{\Gamma}{2}}}{\delta^\Gamma}\exp\left[\frac{\Gamma+2}{2}\right]\left[\kappa_2^2(\tau)(\frac{\bar{\beta}}{\delta^2} + \frac{1}{\tau})\bar{\gamma}_K^{1/2} + \kappa_2^2(\tau)]\bar{\gamma}_K^{1/2}\right.$$

$$+ \frac{\bar{\beta}^{\frac{\Gamma}{2}}}{\delta^\Gamma}\exp\left[\frac{\Gamma+2}{2}\right]\frac{\bar{D}}{\sqrt{\tau}} + 2(\frac{R}{\tau} + \sqrt{d})\sqrt{\delta} \,,$$

*where $C_2(\tau)$ is linear in $\tau^2$, $\kappa_2(\tau) = \max\{\bar{\beta}, T\}\left(\frac{1}{\tau} + \frac{\bar{\beta}}{\delta^2}\right)$.*

**Proof.** The proof of this corollary is almost identical to the proof of Theorem 2. We just need to replace the first bound for the tangent process in Lemma 2 by the second bound. ∎

# D    LEMMAS FOR THE LOGARITHMIC DENSITY

In this section, we introduce auxiliary lemmas to control the gradient and Hessian of the logarithmic density under the manifold hypothesis. Lemma 13, Lemma 14 and Lemma 15 come from Lemma C.1, Lemma C.2, and Lemma C.5 of Bortoli (2022). Since these lemmas do not involve the relationship between $m_t$ and $\sigma_t$, we can directly use the results from Bortoli (2022). Following Bortoli (2022), we also define a empirical version of $q_0$ with $N$ datapoints, i.e. $q_0^N = (1/N)\sum_{k=1}^N X^k$, with $\{X^k\}_{k=1}^N \sim q_0^{\otimes N}$. We denote by $\left(q_t^N\right)_{t>0}$ such that for any $t > 0$ the density w.r.t. the Lebesgue measure of the distribution of $\mathbf{X}_t^N$, and when $N \to +\infty$, $q_t^N = q_t$.

**Lemma 13.** *Assume **Assumption 2**. Then for any $t \in (0, T]$ and $x_t \in \mathbb{R}^d$ we have that*

$$\langle\nabla\log q_t(x_t), x_t\rangle \leq -\|x_t\|^2/\sigma_t^2 + m_R\|x_t\|/\sigma_t^2 \,.$$

*In addition, we have*

$$\|\nabla\log q_t(x_t)\|^2 \leq 2\|x_t\|^2/\sigma_t^4 + 2m_t^2 R^2/\sigma_t^4 \,.$$

**Lemma 14.** *Assume* **Assumption 2**. *Then for any $t \in (0, T]$, $x_t \in \mathbb{R}^d$ and $M \in \mathcal{M}_d \left( \mathbb{R}^d \right)$*

$$\left\langle M, \nabla^2 \log q_t \left( x_t \right) M \right\rangle \leq - \left( 1 - m_t^2 R^2 / \left( 2\sigma_t^2 \right) \right) / \sigma_t^2 \|M\|^2.$$

*In addition, we have*

$$\left\| \nabla^2 \log q_t \left( x_t \right) \right\| \leq \left( 1 + R^2 \right) / \sigma_t^4.$$

The following lemma shows that the derivatives up to the fourth order are uniformly bounded since $\tau \in [T, T^2]$. Thus we can use the stochastic extension of the Alekseev–Gröbner formula (Del Moral and Singh, 2022).

**Lemma 15.** *Assume* **Assumption 2**. *Then, there exists $\bar{C} \geq 0$ such that for any $t \in (0, T]$ we have*

$$\left\| \nabla^2 \log q_t(x) \right\| + \left\| \nabla^3 \log q_t(x) \right\| + \left\| \nabla^4 \log q_t(x) \right\| \leq \bar{C} / \sigma_t^8.$$

The following lemma shows that $\| \partial_t \nabla \log q_t \left( x_t \right) \|$ is bounded. The proof before using the relationship between $\sigma_t$ and $m_t$ is identical compared to Lemma C.3 in Bortoli (2022). For the sake of completeness, we also give the proof process of this part.

**Lemma 16.** *Assume* **Assumption 2**. *Then for any $t \in (0, T]$ and $x_t \in \mathbb{R}^d$ we have*

$$\| \partial_t \nabla \log q_t \left( x_t \right) \| \leq \left( \beta_t / \sigma_t^6 \right) \left( 2 + R^2 \right) \left( R + \|x_t\| \right).$$

**Proof.** Let $N \in \mathbb{N}$ and $t \in (0, T]$. We denote for any $x \in \mathbb{R}^d$, $q_t^N \left( x \right) = \bar{q}_t^N \left( x \right) / \left( 2\pi\sigma_t^2 \right)^{d/2}$ with

$$\bar{q}_t^N \left( x \right) = (1/N) \sum_{k=1}^N e_t^k \left( x \right), \qquad e_t^k(x) = \exp \left[ -\|x - m_t X^k\|^2 / \left( 2\sigma_t^2 \right) \right].$$

Next we denote $f_t^k \triangleq \log e_t^k$. Then we have

$$\partial_t \log \bar{q}_t^N \left( x \right) \sum_{k=1}^N \partial_t f_t^k \left( x \right) e_t^k \left( x \right) / \sum_{k=1}^N e_t^k \left( x \right).$$

Therefore we have

$$\partial_t \nabla \log \bar{q}_t^N (x)$$
$$= \sum_{k=1}^N \partial_t \nabla f_t^k(x) e_t^k(x) / \sum_{k=1}^N e_t^k(x) + \sum_{k=1}^N \partial_t f_t^k(x) \nabla f_t^k(x) e_t^k(x) / \sum_{k=1}^N e_t^k(x)$$
$$- \sum_{k,j=1}^N \partial_t f_t^k(x) \nabla f_t^j(x) e_t^k(x) e_t^j(x) / \sum_{k,j=1}^N e_t^k(x) e_t^j(x)$$
$$= \sum_{k=1}^N \partial_t \nabla f_t^k(x) e_t^k(x) / \sum_{k=1}^N e_t^k(x)$$
$$+ (1/2) \sum_{k,j=1}^N \left( \partial_t f_t^k(x) - \partial_t f_t^j(x) \right) \left( \nabla f_t^k(x) - \nabla f_t^j(x) \right) e_t^k(x) e_t^j(x) / \sum_{k,j=1}^N e_t^k(x) e_t^j(x).$$

In what follows, we provide upper bounds for $|\partial_t f_t^k - \partial_t f_t^j|$, $\|\nabla f_t^k - \nabla f_t^j\|$ and $\partial_t \nabla f_t^k$. First we notice that $\nabla f_t^k(x) = - \left( x - m_t X^k \right) / \sigma_t^2$, and using $m_t \leq 1$ we get

$$\|\nabla f_t^k(x) - \nabla f_t^j(x)\| \leq m_R / \sigma_t^2 \leq R / \sigma_t^2.$$

and

$$\partial_t f_t^k \left( t \right) = \partial_t \sigma_t^2 / \left( 2\sigma_t^4 \right) \|x - m_t X^k\|^2 + \partial_t m_t / \sigma_t^2 \left\langle X^k, x - m_t X^k \right\rangle.$$

Notice the fact that $\partial_t \sigma_t^2 = -2\tau m_t \partial_t m_t = 2\beta_t m_t^2$ and $\partial_t m_t = -\dfrac{\beta_t}{\tau} m_t$, combined with the above equality, we know that

$$\partial_t f_t^k(t) = -\beta_t m_t / \sigma_t^2 \left[ -\left(m_t/\sigma_t^2\right) \|x - m_t X^k\|^2 + \frac{1}{\tau} \left\langle x - m_t X^k, X^k \right\rangle \right]$$

$$= -\beta_t m_t / \sigma_t^2 \left\langle x - m_t X^k, -\left(m_t/\sigma_t^2\right)\left(x - m_t X^k\right) + \frac{1}{\tau} X^k \right\rangle$$

$$= -\beta_t m_t / \sigma_t^4 \left\langle x - m_t X^k, -m_t x + \left(m_t^2 + \frac{\sigma_t^2}{\tau}\right) X^k \right\rangle$$

$$= \beta_t m_t / \sigma_t^4 \left( m_t \|x\|^2 + m_t \left\| X^k \right\|^2 + \left(1 + m_t^2\right) \left\langle x, X^k \right\rangle \right),$$

where the last equality holds that $\tau m_t^2 + \sigma_t^2 = \tau$. The rest of the proof is identical to the Lemma C.3 in Bortoli (2022).

So using $m_t \le 1$ we have

$$\left| \partial_t f_t^k(x) - \partial_t f_t^j(x) \right| \le 2\beta_t m_t^2 R^2 / \sigma_t^4 + \beta_t m_t \left(1 + m_t^2\right) R\|x\|/\sigma_t^4$$

$$\le 2\left(\beta_t/\sigma_t^4\right) R(R + \|x\|)$$

Now we compute $\nabla \partial_t f_t^k(x)$ for any $x \in \mathbb{R}^d$

$$\nabla \partial_t f_t^k(x) = 2\beta_t m_t^2 / \sigma_t^4 x + \left(\beta_t m_t / \sigma_t^4\right) \left(1 + m_t^2\right) X^k.$$

So we can bound the norm of it by

$$\left\| \partial_t \nabla f_t^k(x) \right\| \le 2\left(\beta_t/\sigma_t^4\right)(R + \|x\|).$$

Combining results above we get for any $x \in \mathbb{R}^d$

$$\left\| \partial_t \nabla \log \bar{q}_t^N(x) \right\| \le 2\left(\beta_t/\sigma_t^4\right)(R + \|x\|) + \left(\beta_t/\sigma_t^6\right) R^2(R + \|x\|)$$

$$\le \left(\beta_t/\sigma_t^6\right)\left(2 + R^2\right)(R + \|x\|)$$

Note that

$$\lim_{N \to +\infty} \partial_t \nabla \log q_t^N(x_t) = \partial_t \nabla \log q_t$$

and the proof is complete. ∎

In the following lemma, similar to Chen et al. (2023d), we obtain a better control on the time discretization error instead of controlling $\|\partial_t \nabla \log q_t(x_t)\|$ for $\forall x_t \in \mathbb{R}^d$.

**Lemma 17.** *Assume **Assumption 2** and $X_t$ satisfies the forward process Eq. (1). Define $L = \max_{t \in [0, T-\delta]} \left\| \nabla^2 \log q_{T-t}(\mathbf{Y}_t) \right\| \le \left(1 + R^2\right) / \sigma_\delta^4$, then we have that*

$$\mathbb{E}_{Q_{t_K}^{q_T^\tau}} \left[ \left\| \nabla \ln \frac{q_{T-t_k}}{q_{T-t}}(\mathbf{Y}_{t_k}) \right\|^2 \right]$$

$$\lesssim \tau L^2 d\bar{\gamma}_K + \tau L^2 \bar{\gamma}_K^2 (d\tau + R^2) + \tau L^3 \bar{\gamma}_K^2 + \tau L^4 \bar{\gamma}_K^2 (\beta_T d\bar{\gamma}_K + R^2 \bar{\gamma}_K^2).$$

**Proof.** Due to the property of the forward process, we know that if $S : \mathbb{R}^d \to \mathbb{R}^d$ is the mapping $S(x) := \exp(-(t - t_k))x$, then $q_{T-t_k} = S_{\#} q_{T-t} * \text{normal}\left(0, \tau\left(1 - \exp(-2\int_{t_k}^{t_k+1} \beta_s/\tau ds)\right)\right)$

Similar to Chen et al. (2023d), we define $\alpha = \exp\left[\int_{t_k}^{t_k+1} \frac{\beta_s}{\tau} ds\right] = 1 + O(\bar{\gamma}_K)$ and $\sigma^2 = \tau\left(1 - \exp(-2\int_{t_k}^{t_k+1} \beta_s/\tau ds)\right) = O(\tau \bar{\gamma}_K)$. Then we can use Lemma C.12 of Lee et al. (2022) to obtain

$$\mathbb{E}_{Q_{t_K}^{q_T^\tau}} \left[ \left\| \nabla \ln \frac{q_{T-t_k}}{q_{T-t}}(\mathbf{Y}_{t_k}) \right\|^2 \right]$$

$$\lesssim \tau L^2 d\bar{\gamma}_K + \tau L^2 \bar{\gamma}_K^2 \|\mathbf{Y}_{t_k}\|^2 + \tau L^2 \bar{\gamma}_K^2 \|\nabla \ln q_{T-t}(\mathbf{Y}_{t_k})\|^2$$

$$\lesssim \tau L^2 d\bar{\gamma}_K + \tau L^2 \bar{\gamma}_K^2 (d\tau + R^2) + \tau L^3 \bar{\gamma}_K^2 + \tau L^4 \bar{\gamma}_K^2 (\beta_T d\bar{\gamma}_K + R^2 \bar{\gamma}_K^2).$$

The last inequality follows Lemma 21 and the fact that

$$
\begin{aligned}
\|\nabla \ln q_{T-t}\left(\mathbf{Y}_{t_k}\right)\|^2 &\lesssim \|\nabla \ln q_{T-t}\left(\mathbf{Y}_t\right)\|^2 + \|\nabla \ln q_{T-t}\left(\mathbf{Y}_{t_k}\right) - \nabla \ln q_{T-t}\left(\mathbf{Y}_t\right)\|^2 \\
&\lesssim \|\nabla \ln q_{T-t}\left(\mathbf{Y}_{t_k}\right)\|^2 + L^2(\beta_T d\bar{\gamma}_K + R^2\bar{\gamma}_K^2) \\
&\lesssim L + L^2(\beta_T d\bar{\gamma}_K + R^2\bar{\gamma}_K^2)\,.
\end{aligned}
$$

∎

## E   AUXILIARY LEMMAS

**Lemma 18.** *For any* $s, t \in [0, T]$ *we have*

$$
\int_s^t \beta_{T-u}/\sigma_{T-u}^2 \mathrm{d}u = \left[ -\frac{1}{2}\log\left( \exp\left[ 2\int_0^{T-u} \frac{\beta_v}{\tau}\mathrm{d}v \right] - 1 \right) \right]_s^t,
$$

$$
\int_s^t \beta_{T-u} m_{T-u}^2/\sigma_{T-u}^4 \mathrm{d}u = \left[ (1/2\tau) \Big/ \left( 1 - \exp\left[ -2\int_0^{T-u} \frac{\beta_v}{\tau}\mathrm{d}v \right] \right) \right]_s^t.
$$

**Proof.** We directly compute

$$
\begin{aligned}
\int_s^t \beta_{T-u}/\sigma_{T-u}^2 \mathrm{d}u &= \frac{1}{\tau}\int_s^t \beta_{T-u}\Big/ \left( 1 - \exp\left[ -2\int_0^{T-u} \frac{\beta_v}{\tau}\mathrm{d}v \right] \right) \mathrm{d}u \\
&= \frac{1}{\tau}\int_s^t \beta_{T-u}\exp\left[ 2\int_0^{T-u} \frac{\beta_v}{\tau}\mathrm{d}v \right] \Big/ \left( \exp\left[ 2\int_0^{Tu} \frac{\beta_v}{\tau}\mathrm{d}v \right] - 1 \right) \mathrm{d}u \\
&= -\frac{1}{2}\int_s^t \partial_u \log\left( \exp\left[ 2\int_0^{T-u} \frac{\beta_v}{\tau}\mathrm{d}v \right] - 1 \right) \mathrm{d}u\,.
\end{aligned}
$$

Similarly

$$
\begin{aligned}
&\int_s^t \beta_{T-u} m_{T-u}^2/\sigma_{T-u}^4 \\
&= \frac{1}{\tau^2}\int_s^t \beta_{T-u}\exp\left[ -2\int_0^{T-u} \frac{\beta_v}{\tau}\mathrm{d}v \right] \Big/ \left( 1 - \exp\left[ -2\int_0^{T-u} \frac{\beta_v}{\tau}\mathrm{d}v \right] \right)^2 \mathrm{d}u \\
&= (1/2\tau)\int_t^s \partial_u \left( 1 - \exp\left[ -2\int_0^{T-u} \frac{\beta_v}{\tau}\mathrm{d}v \right] \right)^{-1} \mathrm{d}u.
\end{aligned}
$$

∎

**Lemma 19.** *Assume* **Assumption 1**. *For* $i \in \{1, 2\}$, *we have* $\sigma_{T-t_K}^2(i) \leq 2\delta$ *and* $\sigma_u^{-2}(i) \leq \sigma_{T-t_K}^{-2}(i) \leq \dfrac{1}{\tau} + \dfrac{\bar{\beta}}{\delta^{4-i}}, \forall u \in [T - t_K, T]$.

**Proof.**

$$
\begin{aligned}
\sigma_{T-t_K}^2(i) &= \tau\left( 1 - \exp\left[ -2\int_0^{T-t_K} \frac{\beta_s}{\tau}\,\mathrm{d}s \right] \right) \\
&\leq 2\int_0^{T-t_K} \beta_s\,\mathrm{d}s \leq 2\delta\,,
\end{aligned}
$$

where the first inequality follows from for any $a \geq 0$, $\exp[-a] \geq 1 - a$; the second inequlity follows from **Assumption 1** and $\delta \leq 1$.

$$\sigma_{T-t_K}^{-2}(i) = \frac{1}{\tau} \left( 1 - \exp\left[ -2 \int_0^{T-t_K} \frac{\beta_s}{\tau} \, \mathrm{d}s \right] \right)^{-1} \leq \frac{1}{\tau} \left( 1 + \left( 2 \int_0^{T-t_K} \frac{\beta_s}{\tau} \, \mathrm{d}s \right)^{-1} \right)$$

$$\leq \frac{1}{\tau} + \frac{\bar{\beta}}{\delta^{4-i}} \, ,$$

where the first inequality follows from for any $a \geq 0, 1/(1 + \exp[-a]) \leq 1 + 1/a$, the second inequality follows from **Assumption 1**. It is easy to check that $\sigma_u^{-2}(i) \leq \sigma_{T-t_K}^{-2}(i), \forall u \in [T - t_K, T]$.

∎

Using the bound on $\sigma_{T-t_K}^{-2}(i)$ immediately yields the following control of $\beta_u/\sigma_u^2(i)$.

**Lemma 20.** *Assume* **Assumption 1**. *Then, we have for any* $u \in [T - t_K, T]$*: (1) if* $i = 1$*, then*

$$\frac{\beta_u}{\sigma_u^2(i=1)} \leq \kappa_1(\tau) = \max\{\bar{\beta}, T^2\} \left( \frac{1}{\tau} + \frac{\bar{\beta}}{\delta^3} \right)$$

*(2) if* $i = 2$*, then*

$$\frac{\beta_u}{\sigma_u^2(i=2)} \leq \kappa_2(\tau) = \max\{\bar{\beta}, T\} \left( \frac{1}{\tau} + \frac{\bar{\beta}}{\delta^2} \right)$$

In the rest of this section, we provide the useful lemma to achieve polynomial sample complexity for VE-based models with reverse SDE. As shown in Lemma 13, we also need to control $\mathbb{E}[\|\mathbf{X}_t\|^2]$ in the forward process. The following lemmas shows that this term is bounded by the $R^2$ and exploding variance.

**Lemma 21.** *Suppose that* **Assumption 2** *hold. Let* $(\mathbf{X}_t)_{t \in [0,T]}$ *denote the forward process Eq.* (1). *Then, for all* $t \geq 0$,
$$\mathbb{E}\left[ \|\mathbf{X}_t\|^2 \right] \leq d\sigma_t^2 \vee R^2 \, .$$

**Proof.** As shown in Eq. (2),
$$\mathbb{E}\left[ \|\mathbf{X}_t\|^2 \right] \leq \mathbb{E}\left[ \|\mathbf{X}_0\|^2 \right] + \sigma_t^2 d \leq d\sigma_t^2 \vee R^2 \, .$$

∎

**Lemma 22** (movement bound for VESDE). *Let* $(\mathbf{X}_t)_{t \in [0,T]}$ *denote the forward process Eq.* (1). *For* $0 \leq s < t$ *with* $\delta := t - s$*, if* $\delta \leq 1$*, then*
$$\mathbb{E}\left[ \|\mathbf{X}_t - \mathbf{X}_s\|^2 \right] \lesssim 2\beta_t \delta d + \delta^2 R^2 \, .$$

**Proof.**
$$\mathbb{E}\left[ \|\mathbf{X}_t - \mathbf{X}_s\|^2 \right] \lesssim \mathbb{E}\left[ \left\| \sqrt{2\beta_t} \left( B_t - B_s \right) \right\|^2 \right] + \delta \int_s^t \mathbb{E}\left[ \|\mathbf{X}_r\|^2 \right] \mathrm{d}r \lesssim 2\beta_t \delta d + \delta^2 R^2 \, .$$

∎

Similar to Chen et al. (2023d), we can also show that if we do forward process for time $\delta$, $q_\delta$ will be close to $q_0$ in $W_2$ distance.

**Lemma 23.** *Suppose* **Assumption 2** *holds. Let* $\epsilon_{W_2} > 0$*. If* $\beta_t^2 = t^2$ *and* $\tau = T^2$*, we choose the early stopping parameter* $\delta \leq \frac{\epsilon_{W_2}^{2/3}}{(d+R\sqrt{d})^{1/3}}$*. If* $\beta_t = t$ *and* $\tau = T$*, we choose* $\delta \leq \frac{\epsilon_{W_2}}{(d+R\sqrt{d})^{1/2}}$*. If consider pure VESDE (SMLD) (Eq.* (3)*) with* $\sigma_t^2 = t$*, we choose* $\delta \leq \frac{\epsilon_{W_2}^2}{d}$*. Then we have* $W_2\left( q_\delta, q_0 \right) \leq \epsilon_{W_2}$*.*

**Proof.** For the forward process Eq. (1), we know that $\mathbf{X}_t := m_t\mathbf{X}_0 + \sigma_t Z$, where $Z \sim \mathrm{normal}\,(0, I_d)$ is independent of $X_0$ and $m_t \leq 1$. Hence, for $\delta \lesssim 1$,

$$W_2^2\,(q_0, q_\delta) \leq (1 - m_t)^2 \mathbb{E}\left[\|\mathbf{X}_0\|^2\right] + \mathbb{E}\left[\|\sigma_\delta Z\|^2\right].$$

For $\beta_t = t^2$ and $\tau = T^2$, we have that

$$W_2^2\,(q_0, q_\delta) \leq \delta^3 d + \frac{R^2\delta^6}{T^2}$$

Hence, we can take $\delta \leq \frac{\epsilon_{W_2}^{2/3}}{(d + R\sqrt{d})^{1/3}}$. For $\beta_t = t$ and $\tau = T$, we have that

$$W_2^2\,(q_0, q_\delta) \leq \delta^2 d + \frac{R^2\delta^4}{T^2}$$

Hence, we can take $\delta \leq \frac{\epsilon_{W_2}}{(d + R\sqrt{d})^{1/2}}$. For pure VESDE (Eq. (3)) with $\sigma_t = t$, we have

$$W_2^2\,(q_0, q_\delta) \leq \delta d.$$

∎

## F ADDITIONAL SYNTHETIC EXPERIMENTS

In this section, we do synthetic experiments to show the power of our new forward process with small drift term in different setting.

### F.1 THE SYNTHETIC EXPERIMENTS WITH ACCURATE SCORE FUNCTION

In this section, we do numerical experiments on 2-dimension Gaussian distribution to show the power of our new VESDE forward process in balancing different error sources.

**Experiment Setting.** We set the mean of target distribution $\mathbb{E}[q_0] = [6, 8]$, the covariance matrix $\mathrm{Cov}[q_0] = \begin{bmatrix} 25 & 5 \\ 5 & 4 \end{bmatrix}$, the diffusion time $T = 2$, $\tau = T^2$ and the reverse beginning distribution is $\mathcal{N}(0, T^2\mathbf{I})$. We choose uniform stepsize $\gamma_k = h, \forall k \in [K]$ where $h \in \{0.005, 0.01, 0.02, 0.04\}$. For score functions, we directly calculate the ground truth score function instead of learning it by the score matching objective. We calculate the KL divergence between the generation distribution and target distribution $q_0$ as the experiments.

**The implementable algorithm.** We choose three different VESDE forward processes in the experiments: (1) aggressive $\beta_t = t^2$ with $\tau = T^2$; (2) conservative $\beta_t = t$ with $\tau = T^2$ and (3) VESDE without drift term Eq. (3) with $\sigma_t^2 = t^2$. After determining the forward process, we run the reverse SDE with the above $\gamma_k, k \in [K]$. For the discretization scheme, we choose two common method: exponential integrator (EI) (Zhang and Chen, 2022) and Euler-Maruyama (EM) discretization (Ho et al., 2020).

**Observations.** The experimental results are shown in Fig. 1. We note that the red line (EI, VESDE without drift, $\sigma_t^2 = t^2$) and orange line (conservative drift VESDE, $\beta_t = t$ and $\tau = T^2$) has a similar trend. Furthermore, the conservative drift VESDE has better performance compared to pure VESDE without drift term. Hence, our new forward process is representative enough to represent current VESDE, as discussed in Section 3.1.

The experimental results also support our theoretical results and show the power of the new forward process in balancing different error terms. As shown in Fig. 1, the process with aggressive $\beta_t = t^2$ with small drift term achieves the best and second performance in EI and EM discretization since it can balance the reverse beginning and discretization. The third best process is conservative $\beta_t = t$ with the small drift term. The reason is that though it can not achieve a $\exp(-T)$ forward process guarantee, it also has a constant decay on prior information, as shown in Section 3.1. This decay slightly reduces the effect of the reverse beginning error. The worse process is VESDE without drift term since it is hard to balance different error sources. Our experimental results also show that EI discretization is better than EM discretization.

### F.2 THE SYNTHETIC EXPERIMENTS WITH APPROXIMATED SCORE FUNCTION

In this section, instead of using an accurate score function, we train an approximated score function on the pure VESDE (Eq. (3)) without drift term on two synthetic datasets: multiple Swiss rolls and 1-D GMM. Then, for the drift VESDE, we do not train the approximated score corresponding to Eq. (1); we directly use the approximated score learned by pure VESDE and show that the drift VESDE can improve the generated distribution without the training process.

**Datasets.** The 1-D GMM distribution contains three modes:

$$\frac{3}{10}\mathcal{N}\left(-8, 0.01\right) + \frac{3}{10}\mathcal{N}\left(-4, 0.01\right) + \frac{4}{10}\mathcal{N}\left(3, 1\right).$$

For multiple Swiss rolls, we use a similar code compared to Listing 2 of Lai et al. (2023), except Line 6. We change Line 6. to data /=10. to obtain a larger variance dataset. Each dataset contains 50000 datapoints.

**The implementable algorithm.** In this subsection, we adapt the code of a particular repository[1], which corresponds to Eq. (3) with $\sigma_t^2 = t$, as mentioned in the Appendix C.2 of Karras et al. (2022). Hence, we choose two forward processes: (1) conservative $\beta_t = 1$ with $\tau = T$; (2) pure VESDE without drift term (Eq. (3)) with $\sigma_t^2 = t$. To match our analysis, we choose two sampling methods for the reverse process: Euler-Maruyama method for reverse SDE and RK45 ODE solver for the reverse PFODE method.

We note that although aggressive setting $\beta_t = t$ and $\tau = T$ has shown its power in theory (Lemma 5) and the experiments with accurate score (Fig. 1), other sampling issues may arise in practice. We leave the experimental exploration for drift VESDE with aggressive $\beta_t$ as a future work.

**The training detail.** For each dataset, we train a score function with pure VESDE (Eq. (3), $\sigma_t^2 = t$) by using exactly the same network compared to the above repository. We train for 200 epochs with batch size 200 and learning rate $10^{-4}$. For both training and inference, the start time is $\delta = 10^{-5}$. For the conservative VESDE, we directly adapt the checkpoint learned by the pure VESDE since the conservative drift VESDE has a similar trend compared to pure VESDE, as shown in Fig. 1.

**Observation.** We do experiments with $T = 100$ and lager $T = 625$ and these two choice show similar phenomenon. In this paragraph, we first use $T = 100$ as an example to discuss the results. As shown in Table 1, the conservative drift VESDE has smaller KL divergence compared to pure VESDE under all sampling methods and datasets. From Fig. 2 and Fig. 3, it is clear that pure VESDE has low density on the Swiss roll except the center one, which means that though pure VESDE can deal with small $\mathbb{E}[q_0]$, it is hard to deal with large dataset variance $\mathrm{Cov}[q_0]$, as we discuss in Section 4. For conservative drift VESDE ($\beta_t = 1$ and $\tau = T$), as we discuss in Section 3.1, there is a constant decay on the prior information $\mathbb{E}[q_0]$ and $\mathrm{Cov}[q_0]$, which is helpful in deal with large dataset mean and variance. The experimental results support our augmentation. Fig. 2c, Fig. 3c and Fig. 4c show that the density of the generated distribution is more uniform compared to pure VESDE, which means that the drift VESDE can deal with large dataset mean and variance.

We also do experiments with larger $T = 625$. As we discuss in Section 4, larger $T$ will reduce the influence of the prior data information and have greater generated distribution, as shown in Fig. 4b and Fig. 3c. The experiments of 1D-GMM (Fig. 4) show a similar phenomenon compared to the multi Swiss rolls.

---

[1]https://colab.research.google.com/drive/120kYYBOVa1i0TD85RjlEkFjaWDxSFUx3?usp=sharing

Table 1: The KL divergence for pure VESDE (Eq. (3)) and conservative drift VESDE with different sampling method.

| Forward Process | 1-D GMM | | Swiss roll | |
|---|---|---|---|---|
| | Reverse SDE | PFODE | Reverse SDE | PFODE |
| Pure VESDE ($T = 100$) | 0.082 | 0.434 | 9.58 | 21.05 |
| Drift VESDE ($T = 100$) | 0.043 | 0.249 | 8.71 | 7.77 |
| Pure VESDE ($T = 625$) | 0.027 | 0.057 | 8.00 | 8.20 |
| Drift VESDE ($T = 625$) | **0.025** | 0.031 | 7.95 | **7.21** |

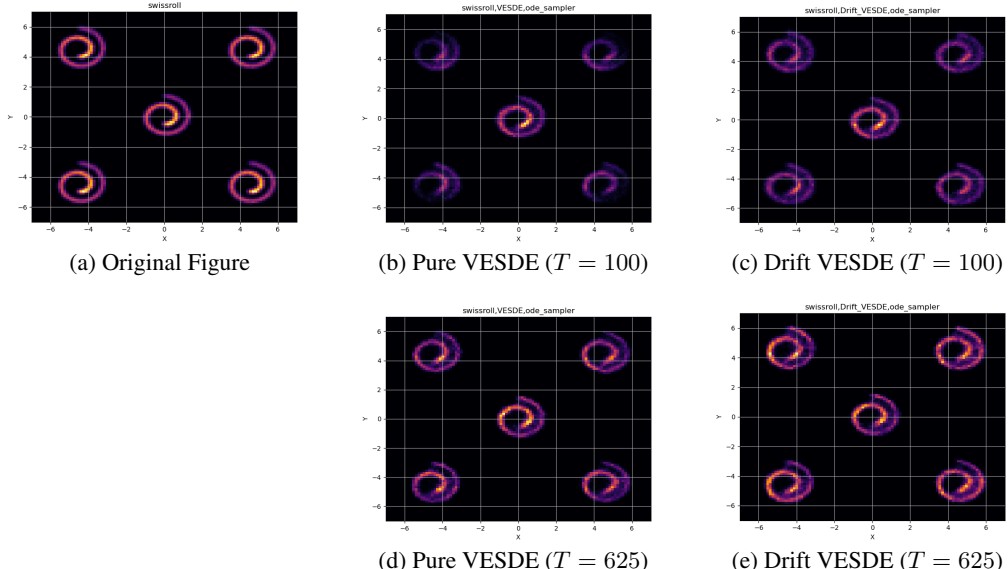

(a) Original Figure     (b) Pure VESDE ($T = 100$)     (c) Drift VESDE ($T = 100$)

(d) Pure VESDE ($T = 625$)     (e) Drift VESDE ($T = 625$)

Figure 3: Experiment results of Swiss roll with reverse PFODE

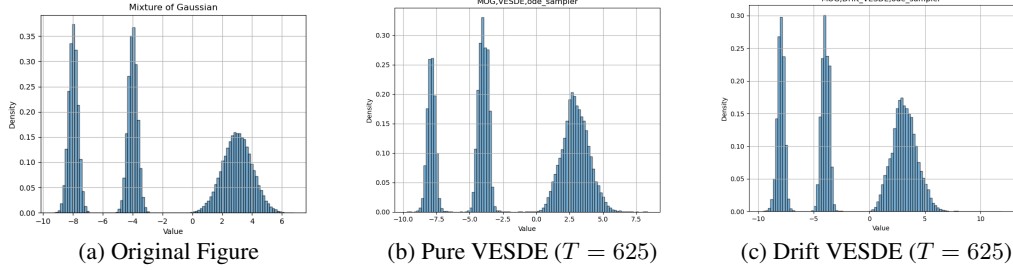

(a) Original Figure     (b) Pure VESDE ($T = 625$)     (c) Drift VESDE ($T = 625$)

Figure 4: Experiment results of 1D-GMM with reverse PFODE