# OpenReview forum: "The Convergence of Variance Exploding Diffusion Models under the Manifold Hypothesis"
_ICLR.cc/2024/Conference — Submitted to ICLR 2024_

### Official Review · Reviewer_A2uB · 2023-10-31

**Soundness:** 3 good
**Presentation:** 3 good
**Contribution:** 3 good
**Rating:** 6
**Confidence:** 4

**Summary:**

The paper proposes and studies the convergence of a variance-exploding diffusion model by introducing a small drift term and manifold hypothesis, and proves a faster exponential rate. The paper is mainly theoretical, and also includes numerical experiments to corroborate the theory.

**Strengths:**

The paper proposes a new variance exploding diffusion model, and proves under the manifold hypothesis that the underlying algorithm converges faster. The paper is mostly well written, and I enjoyed reading it. I have also checked part of the proofs, and they seem to be correct.

**Weaknesses:**

There are several weaknesses of this paper:

(1) It was claimed in the introduction that the underlying algorithm converges at rate $\exp(-T)$. However, upon looking at Theorem 2, there is a term which is linear in $T$ (under the Wasserstein distance). The authors may comment and explain this "discrepancy" -- otherwise it seems to be that the abstract is exaggerated.

(2) As the authors pointed out themselves, the analysis is similar to De Bortoli (2022). I understand that the technicality comes from analyzing the "tangent" process to avoid $\exp(T)$ term. The authors may highlight/summary the novelty earlier in the introduction.

(3) In most theorems/corollaries, the authors distinguish "aggressive" $\beta_t = t^2$, and "conservative" $\beta_t$'s (e.g. $\beta_t = t$). I wonder if there is some "phase transition" at exponent $2$ (i.e. $\beta_t = t^\alpha$ with $\alpha < 2$ and $\alpha = 2$). The authors may comment on this point.

(4) Now all the experiments are deferred to appendix. I feel that the authors may lift some of experiments to the main text. (and put some proof techniques to appendix.)

**Questions:**

See the weaknesses.

---

> ### Author Response · Authors · 2023-11-17
>
> Thank you for your valuable comments and suggestions. We provide our response to each question below.
>
> **Weakness 1. The discussion on the faster forward convergence $\exp{(-T)}$ and $\operatorname{Poly}(T)\\bar{\gamma}_K$ term.**
>
> When discussing the faster convergence rate of our new forward process, we focus on the reverse beginning error term, which corresponds to the forward process, instead of the discretization error term. As shown in Section 5, a faster forward convergence guarantee is important to balance different error sources and achieve great sample complexity.
>
> For the $\operatorname{Poly}(T)\bar{\gamma}_K$ term, which corresponds to the discretization scheme and smallest stepsize $\bar{\gamma}_K$, we can choose small enough stepsize to eliminate the influence of $\operatorname{Poly}(T)$. Furthermore, $\operatorname{Poly}(T)\bar{\gamma}_K$ term for the discretization terms is widely appeared in the literature (Theorem 1 of [1] under the  Wasserstein distance; Theorem 2 of [2] under the TV distance). We note that under the choice of $T$ and $\bar{\gamma}_K$ in Corollary 2 of [1], the first discretization term of Theorem 1 in [1] depends on $T^2 \sqrt{\bar{\gamma}_K}$.
>
> We have now made the definition of the faster convergence rate clearer in the abstract in the revision of our paper (highlighted in green).
>
> **Weakness 2. The technique novelty.**
>
> Thanks for your helpful comments. We have now summarized the technique novelty for controlling the tangent process to avoid the $\exp(T)$ term by using the exploding variance property of VESDE at the end of Section 1  in the revision of our paper (highlighted in green).
>
> **Weakness 3. Transition situation $\beta_t = t^{\alpha_1}, \alpha_1\in [1,2]$ between conservative $t$ and the most aggressive $t^2$.**
>
> We use $\tau  = T^2$ as an example and discuss the influence of different $\beta_t =t^{\alpha_1}, \alpha_1\in [1,2]$ for the forward and reverse process. For the forward process, the "phase transition" appears at $1+\ln(T-\ln(T))/\ln(T)$. When $\alpha_1 \geq 1+\ln(T-\ln(T))/\ln(T)$ is aggressive, the forward convergence rate is faster than $\exp(-T)$, which means the influence of $T$ is logarithmic factors and achieve great balance in reverse SDE (Section 5). For the analysis of the unified tangent-based framework, as discussed in section 6, whether $\alpha_1$ can be aggressive depends on the form of the reverse process.
>
> We have now added more details and discussion about the different phases of $\alpha_1$ in Section 4 in the revision of our paper (highlighted in green).
>
> **Weakness 4: The additional synthetic experiments..**
>
> Thanks for your helpful comments. We use pure VESDE as a baseline and add more synthetic experiments in the main text to show the advantage of our new forward process with aggressive and conservative drift terms. More specifically, For aggressive drift VESDE, we show that the new process balances the different error sources. For conservative drift VESDE, we further consider the approximated score function and show that the conservative drift VESDE can improve the quality of generated distribution without training. More specifically, the conservative drift VESDE directly uses the models training by pure VESDE and reduces the influence of the large data mean and variance to obtain better images. We verify this phenomenon in different sampling methods (Euler-Maruyama method for reverse SDE and RK45 for PFODE), different $T$, and different synthetic datasets. We also calculate the KL divergence between the generated distribution and the original distribution to support our augmentation.
>
> We have now added more experiment results and discussion about the advantages of our new process in Section 7 and Appendix F in the revision of our paper (highlighted in green).
>
>
>
> [1] Valentin De Bortoli. Convergence of denoising diffusion models under the manifold hypothesis. Trans. Mach. Learn. Res., 2022, 2022. URL https://openreview.net/forum?id=MhK5aXo3gB.
>
> [2] Sitan Chen, Sinho Chewi, Jerry Li, Yuanzhi Li, Adil Salim, and Anru Zhang. Sampling is as easy as learning the score: theory for diffusion models with minimal data assumptions. In The Eleventh International Conference on Learning Representations, ICLR 2023, Kigali, Rwanda, May 1-5, 2023. OpenReview.net, 2023d. URL https://openreview.net/pdf?id=zyLVMgsZ0U_.

---

> ### Author Response · Authors · 2023-11-20
>
> Dear Reviewer A2uB:
>
> We thank you once again for your careful reading of our paper and your constructive comments and suggestions. We will appreciate it very much if you could let us know whether all your concerns are addressed. We are also more than happy to answer any further questions in the remaining discussion period.
>
> Best, Authors.

---

> > ### Comment · Reviewer_A2uB · 2023-11-21
> >
> > I thank the authors for the detailed response. The score is increased to 6.

---

> > > ### Author Response · Authors · 2023-11-22
> > >
> > > Thank you for your positive feedback and support for our work! We are also more than happy to answer any further questions.

---

### Official Review · Reviewer_HKJt · 2023-10-31

**Soundness:** 3 good
**Presentation:** 3 good
**Contribution:** 3 good
**Rating:** 6
**Confidence:** 3

**Summary:**

The forward process, which has been the state-of-the-art in many tasks, is Variance Exploding based diffusion models. However, unlike VPSDE, it had the drawback of a slow convergence rate. Consequently, this paper introduces a new VESDE that exhibits a faster convergence rate. Additionally, it demonstrated the first polynomial sample complexity based on the realistic manifold hypothesis. Experimentally, it showed quantitative convergence guarantees for the state-of-the-art under such settings.

**Strengths:**

1. The introduction of a new forward VESDE process with an unbounded diffusion coefficient and a small drift term, allowing for faster convergence, is a valuable contribution.

2. The paper leverages the manifold hypothesis to achieve a polynomial sample complexity, providing practical insights for real-world applications.

3. The proposal of a unified tangent-based framework for analyzing VE-based models with probability flow ODE is a novel approach to address the convergence guarantee.

**Weaknesses:**

1. While the paper introduces new concepts and theoretical results, it would have been beneficial to have more specific experiments on datasets.

2. While it's understood that the introduced VESDE exhibits a faster convergence rate than VPSDE, there's no guarantee that it offers better fidelity.

3. I find it puzzling whether the variance-exploding property has any utility beyond qualitative convergence guarantees.

**Questions:**

Please refer to Weaknesses

---

> ### Author Response · Authors · 2023-11-17
>
> Thank you for your valuable comments and suggestions. We provide our response to each question below.
>
> **Weakness 1: The additional synthetic experiments.**
>
> We use pure VESDE as a baseline and add more synthetic experiments in the main text to show the advantage of our new forward process with aggressive and conservative drift terms. More specifically, For aggressive drift VESDE, we show that the new process balances the different error sources. For conservative drift VESDE, we further consider the approximated score function and show that the conservative drift VESDE can improve the quality of generated distribution without training. More specifically, the conservative drift VESDE directly uses the models training by pure VESDE and reduces the influence of the large data mean and variance to obtain better images. We verify this phenomenon in different sampling methods (Euler-Maruyama method for reverse SDE and RK45 for PFODE), different $T$, and different synthetic datasets. We also calculate the KL divergence between the generated distribution and the original distribution to support our augmentation.
>
> We have now added more experiment results and discussion about the advantages of our new process in Section 7 and Appendix F in the revision of our paper (highlighted in green).
>
> **Weakness 2: The utility of the variance-exploding property.**
>
> As shown in Figure 3 of [1], the VESDE (Eq (3) in our paper) with $\sigma_t^2 = t^2$ has great PFODE curvature compared to VPSDE and VESDE with $\sigma_t^2 = t$, which relies have on the large variance term $\sigma_t^2 =t^2$ and $q_T = \mathcal{N}(0, T^2 \mathbf{I})$. Based on this PFODE curvature, VESDE with $\sigma_t^2 = t^2$ performs well in multi-step [2] and one-step [3] image generation. Furthermore, for our new drift VESDE, we show that it performs better than pure $\sigma_t^2=t^2$ or  $\sigma_t^2=t$ on the synthetic experiments since it can balance different error sources.
>
> **Weakness 3: The better fidelity with drift VESDE.**
>
> The above discussion shows that drift VESDE generated distributions closer to the original distribution in KL divergence on the synthetic experiments. Then, we explain why drift VESDE has better fidelity than pure VESDE from the theoretical perspective.
>
> Before comparing drift VESDE and previous pure VESDE, we update Corollary 1 by more careful calculation on the time discretization error (Lemma 17) and the early stopping parameter (Lemma 23). With this result, we show that the faster forward convergence rate allows balance in three error sources and finally leads to better sample complexity compared to the previous guarantee for pure VESDE [4]. The results show that the drift VESDE can use smaller steps to achieve the same accuracy, which means this model can use the same steps to achieve better fidelity compared to pure VESDE.
>
> We have now added a detailed comparison and discussion on the sample complexity of pure VESDE in Remark 1 in the revision of our paper (highlighted in green).
>
>
>
> [1] Tero Karras, Miika Aittala, Timo Aila, and Samuli Laine. Elucidating the design space of diffusion-based generative models. Advances in Neural Information Processing Systems, 35:26565–26577, 2022.
>
> [2] Kim, D., Kim, Y., Kang, W., & Moon, I. C. (2022). Refining generative process with discriminator guidance in score-based diffusion models. *arXiv preprint arXiv:2211.17091*.
>
> [3] Yang Song, Prafulla Dhariwal, Mark Chen, and Ilya Sutskever. Consistency models. ICML 2023,
>
> [4] Holden Lee, Jianfeng Lu, and Yixin Tan. Convergence for score-based generative modeling with polynomial complexity. Advances in Neural Information Processing Systems, 35:22870–22882, 2022.

---

> ### Author Response · Authors · 2023-11-20
>
> Dear Reviewer HKJt:
>
> We thank you once again for your careful reading of our paper and your constructive comments and suggestions. We will appreciate it very much if you could let us know whether all your concerns are addressed. We are also more than happy to answer any further questions in the remaining discussion period.
>
> Best, Authors.

---

> ### Author Response · Authors · 2023-11-22
>
> Dear Reviewer HKJt,
>
> We thank you again for your tremendously valuable review of our paper. As the discussion stage is coming to an end, please let us know if you have any questions about our responses or any further concerns.
>
> We would be grateful to answer any further questions in the remaining discussion period.
>
> Best, Authors.

---

> > ### Comment · Reviewer_HKJt · 2023-11-22
> >
> > The authors had addressed my questions, so I raised my initial rating.

---

> > > ### Author Response · Authors · 2023-11-22
> > >
> > > Thank you for your positive feedback and support for our work! We are also more than happy to answer any further questions.

---

### Official Review · Reviewer_U2Hx · 2023-10-31

**Soundness:** 3 good
**Presentation:** 3 good
**Contribution:** 3 good
**Rating:** 6
**Confidence:** 3

**Summary:**

Score-based diffusion models have gained wide attention in recent years. Two common SDEs used are variance exploding SDE (VESDE) and variance preserving SDE. This paper proposes a new forward VESDE. Unlike the previous theoretical works that assuming constant diffusion coefficient, the paper allows unbounded coefficient that grows in time, which is closer to SOTA VE-based models. The new forward process allows a faster exponential convergence rate. The polynomial sample complexity is obtained for VE-based models with reverse SDE under the manifold hypothesis. The paper then proposes the unified tangent-based analysis framework for VE-based models, and the first quantitative guarantee for SOTA VE-based models with reverse PFODE is obtained.

**Strengths:**

The paper is relatively well written. The model, and the contributions are well stated. With the new VESDE forward process, the polynomial sample complexity is obtained for VE-based models with reverse SDE under the manifold hypothesis. Moreover, the first quantitative guarantee for SOTA VE-based models with reverse PFODE is obtained.

**Weaknesses:**

The forward process the paper proposes, to me, is just a small modification of variance preserving SDE with a parameter $\tau$. When $\tau=1$, it is exactly the well-studied variance preserving SDE in the literature. Because of that, it is not clear to me the advantage of proposing this new VESDE given that it seems to be quite similar to the VPSDE.

Moreover, since it is a small extension of the VPSDE, the paper should be more transparent about the technical contributions. What is the technical novelty and difficulty to extend the existing theoretical works in this field to allow this added parameter $\tau$. For example, Lemma 13, Lemma 14, Lemma 15 all come from De Bortolli (2022). The authors should make it more clear what is the novelty that arises from this new VESDE.

It is not clear the advantage of this new VESDE compared to VPSDE or other diffusion models. It would be helpful if the authors can compare the complexity with the existing literature to demonstrate theoretically that this new VESDE can indeed outperform or at least comparable to the existing models. What is more important is the paper lacks convincing numerical experiments. There is only a very small numerical section in the appendix about a 2-dimensional Gaussian distribution, which is not enough to suggest this new VESDE model is promising. Given that this VESDE is just VPSDE with an added parameter $\tau$, would it be possible for the authors to utilize the publicly available codes from the previous literature to see if it works well? Also, in terms of practice, it seems that it is quite often in the literature people simply take $T=1$. If that is the case, then you get $\tau=1$ which corresponds exactly to the VPSDE. The point is that without numerical experiments, it is not clear to me why this new VESDE is a good idea, and what advantage it can bring compared to VPSDE.

**Questions:**

In the abstract, please mention what T stands for so the readers who are not familiar with this field can understand it better.

On page 1, in the last paragraph, “the these models” is a typo.

On page 1, the description that “The VESDE corresponds to a Brownian motion” is not one hundred percent accurate in the sense that my understanding is that VESDE has a deterministic and non-trivial diffusion term in front of the Brownian motion. As a result, more rigorously speaking, VESDE is a Brownian motion with a (deterministic) time change.

On page 2, “We propose a new forward VESDE with the unbounded coefficient $\beta_{t}$ and a small drift term.” It is not clear to me whether this description is accurate. Maybe it is better to say “and a drift term that is typically small”? The reason is that in your model, you can choose $\beta_{t}=t^{2}$ and $\tau=T$. Under this choice, when $t$ is of the order $T$, the drift term can be large?

On page 6, it would be better if you can add some more detail about Assumption 3. Currently, you are saying “similar to (Chen et al. 2022; Benton et al. 2023)”, is your Assumption 3 exactly the same as in (Chen et al. 2022; Benton et al. 2023)? If yes, you can simply make it more transparent. If not, please comment on the difference.

On page 7, “We note that Theorem 2 has exponential on $R$ and $\delta$” did you mean “exponential dependence”?

---

> ### Author Response · Authors · 2023-11-17
>
> Thank you for your valuable comments and suggestions. We provide our response to each question below.
>
> **Weakness 1: The advantage of the new drift VESDE in the application.**
>
> We note that pure VESDE (eq (3) in our paper) achieves SOTA performance in multi-step [1] and one-step image generation [2]. Hence, we use pure VESDE as a baseline and add more synthetic experiments in the main text to show the advantage of our new forward process with aggressive and conservative drift terms. More specifically, For aggressive drift VESDE, we show that the new process balances the different error sources. For conservative drift VESDE, we further consider the approximated score function and show that the conservative drift VESDE can improve the quality of generated distribution without training. More specifically, the conservative drift VESDE directly uses the models training by pure VESDE and reduces the influence of the large data mean and variance to obtain better images. We verify this phenomenon in different sampling methods (Euler-Maruyama method for reverse SDE and RK45 for PFODE), different $T$, and different synthetic datasets. We also calculate the KL divergence between the generated distribution and the original distribution to support our augmentation.
>
> We have now added more experiment results and discussion about the advantages of our new process in Section 7 and Appendix F in the revision of our paper (highlighted in green).
>
> **Weakness 2: The advantage of the new drift VESDE in the theory.**
>
> We note [3] achieve the polynomial sample complexity for pure VESDE with $\sigma_t^2 =t$ and reverse SDE under strong LSI and unrealistic score assumptions. By more careful calculation on the time discretization error (Lemma 17) and the early stopping parameter (Lemma 23), we update Corollary 1, which has a better dependence on $\epsilon_{W_2}$. With this result, we show that the drift VESDE with reverse SDE has a better sample complexity compared to the current sample complexity for pure VESDE under the realistic assumption. This result is also supported by our synthetic experiments (Figure 1).
>
> We have added a detailed comparison and discussion on the sample complexity of pure VESDE in Remark 1 in the revision of our paper (highlighted in green).
>
> **Weakness 3: The technique novelty.**
>
> The key step of our unified tangent-based framework is the tangent process lemma (Lemma 2). We note that when considering reverse PFODE, the original tangent process lemma in [3] even introduces an additional $\exp(T)$ in the discretization error when considering VPSDE. For VESDE with PFODE, we need to carefully control the tangent process by using the exploding variance of VESDE to avoid $\exp(T)$ in the discretization error.
>
> We have summarized the technique novelty at the end of Section 1 in the revision paper (highlighted in green).
>
> **Weakness 4: The choice of $T$ in the VESDE setting.**
>
> We note that for the VESDE setting, previous works usually choose large $T$ to obtain large variance $\mathcal{N}(0, \sigma_T^2)$ and achieve good performance. For example, [1] choose $T=80$ for pure VESDE. Furthermore, we also show that with larger $T$, our drift VESDE performs better on the synthetic experiments (Figure 3 and Table 1 in Appendix F).
>
> Thanks for pointing out the typos in the question part. We have corrected them in the revision paper (highlighted in green).
>
> [1] Kim, D., Kim, Y., Kang, W., & Moon, I. C. (2022). Refining generative process with discriminator guidance in score-based diffusion models. *arXiv preprint arXiv:2211.17091*.
>
> [2] Yang Song, Prafulla Dhariwal, Mark Chen, and Ilya Sutskever. Consistency models. ICML 2023,
>
> [3] Valentin De Bortoli. Convergence of denoising diffusion models under the manifold hypothesis. Trans. Mach. Learn. Res., 2022, 2022. URL https://openreview.net/forum?id=MhK5aXo3gB.

---

> > ### Comment · Reviewer_U2Hx · 2023-11-22
> >
> > Thanks for carefully addressing all the issues raised in the previous round. I have raised the score.

---

> > > ### Author Response · Authors · 2023-11-23
> > >
> > > Thank you for your positive feedback and support for our work! We are also more than happy to answer any further questions.

---

> ### Author Response · Authors · 2023-11-20
>
> Dear Reviewer U2Hx:
>
> We thank you once again for your careful reading of our paper and your constructive comments and suggestions. We will appreciate it very much if you could let us know whether all your concerns are addressed. We are also more than happy to answer any further questions in the remaining discussion period.
>
> Best, Authors.

---

> ### Author Response · Authors · 2023-11-22
>
> Dear Reviewer U2Hx,
>
> We thank you again for your tremendously valuable review of our paper. As the discussion stage is coming to an end, please let us know if you have any questions about our responses or any further concerns.
>
> We would be grateful to answer any further questions in the remaining discussion period.
>
> Best, Authors.

---

### Official Review · Reviewer_byF9 · 2023-11-01

**Soundness:** 3 good
**Presentation:** 3 good
**Contribution:** 3 good
**Rating:** 8
**Confidence:** 2

**Summary:**

This paper introduces a new forward Variance-Exploding stochastic differential equation (VESDE) process by including a drift term whose coefficient can be unbounded. The resulting VPSDE process that enjoys the faster $\exp(-T)$ convergence rate and the authors establish the first polynomial sample complexity result under a manifold hypothesis. The authors also propose a unified framework for VE-based models and obtain the first quantitative convergence results for VE models with probability flow ODE.

**Strengths:**

The paper introduces a drift term to balance out the beginning, discretization and score function errors, which allows the authors to establish faster forward convergence rate and a polynomial sample complexity for the reversed process. The newly proposed VESDE process could be of practical relevance as demonstrated on a toy 2-D example.

**Weaknesses:**

I'm wondering how important the manifold assumption is. Some discussion on this may be helpful.

**Questions:**

Is there a reason for using different metrics in Sections 5 (total variation) and 6 (Wasserstein)?

---

> ### Author Response · Authors · 2023-11-17
>
> Thank you for your valuable comments and suggestions. We provide our response to each question below.
>
> **Weakness 1: The importance of the Manifold Hypothesis.**
>
> We discuss the manifold hypothesis from the empirical and theoretical perspectives. From the empirical perspective, this assumption is supported by much empirical evidence [1] and is naturally satisfied by the image datasets. From the theoretical perspective, this assumption is more realistic compared to the Lipschitz score assumption since it allows the expansion phenomenon of the score in the application. Hence, many current theoretical works analyzing the VPSDE setting also consider the manifold hypothesis \[2\]\[3\].
>
> We have now added more discussion on the manifold hypothesis in Section 4 in the revision of our paper (highlighted in green).
>
> **Question 1. The discussion on the TV+Wasserstein and pure Wasserstein guarantee.**
>
> As discussed in Section 3.1 of [4], since the manifold hypothesis encompasses distributions that admit a continuous density on a lower dimensional manifold, which will lower bound the TV or KL divergence by a constant, studying TV distance or KL divergence bound will vacuous results.
>
> Hence, there are two types of guarantee for VESDE with reverse SDE. The first one is the TV+Wasserstein guarantee, which is used by VPSDE \[2\]\[3\]. This guarantee relies on the early stopping technique and divides the guarantee into two phases. In the first phase $t\in [0, T-\delta]$, the score function is well-defined, and we use Girsanov's Theorem to obtain the TV guarantee in this phase. For the second phase $t\in [T-\delta, T]$, we choose the appropriate early stopping parameter $\delta$ to obtain a $W_2$ close guarantee instead of using the exploding score, which is not well defined when $t=T$. The second one is the pure Wasserstein guarantee, which is used by VPSDE with tangent-based analysis [4].
>
> For reverse PFODE, since the reverse process lacks randomness, we can not use Girsanov's theorem (the above first guarantee). Hence, we provide a pure Wasserstein guarantee for reverse PFODE. Recently, some works \[5\]\[6\]\[7\] achieve TV + Wasserstein guarantee for VPSDE with reverse PFODE. However, they add additional corrector components [5] or have additional assumptions on the discretization process \[6\]\[7\]. We leave the TV+Wasserstein guarantee for VESDE with reverse PFODE as an interesting future work.
>
>
>
> [1] Phillip Pope, Chen Zhu, Ahmed Abdelkader, Micah Goldblum, and Tom Goldstein. The intrinsic dimension of images and its impact on learning. arXiv preprint arXiv:2104.08894, 2021.
>
> [2] Sitan Chen, Sinho Chewi, Jerry Li, Yuanzhi Li, Adil Salim, and Anru Zhang. Sampling is as easy as learning the score: theory for diffusion models with minimal data assumptions. ICLR 2023. OpenReview.net, 2023d. URL https://openreview.net/pdf?id=zyLVMgsZ0U_.
>
> [3] Holden Lee, Jianfeng Lu, and Yixin Tan. Convergence of score-based generative modeling for general data distributions. In International Conference on Algorithmic Learning Theory, pages 946–985. PMLR, 2023.
>
> [4] Valentin De Bortoli. Convergence of denoising diffusion models under the manifold hypothesis. Trans. Mach. Learn. Res., 2022, 2022. URL https://openreview.net/forum?id=MhK5aXo3gB.
>
> [5] Sitan Chen, Sinho Chewi, Holden Lee, Yuanzhi Li, Jianfeng Lu, and Adil Salim. The probability flow ode is
> provably fast. arXiv preprint arXiv:2305.11798, 2023c
>
> [6] Sitan Chen, Giannis Daras, and Alex Dimakis. Restoration-degradation beyond linear diffusions: A non-
> asymptotic analysis for ddim-type samplers. ICML 2023.
>
> [7] Li, G., Wei, Y., Chen, Y., & Chi, Y. (2023). Towards Faster Non-Asymptotic Convergence for Diffusion-Based Generative Models. *arXiv preprint arXiv:2306.09251*.

---

> > ### Comment · Reviewer_byF9 · 2023-11-22
> >
> > I thank the authors for the detailed response which have addressed my questions. I have increased the score to 8.

---

> > > ### Author Response · Authors · 2023-11-22
> > >
> > > Thank you for your positive feedback and support for our work! We are also more than happy to answer any further questions.

---

> ### Author Response · Authors · 2023-11-20
>
> Dear Reviewer byF9:
>
> We thank you once again for your careful reading of our paper and your constructive comments and suggestions. We will appreciate it very much if you could let us know whether all your concerns are addressed. We are also more than happy to answer any further questions in the remaining discussion period.
>
> Best, Authors.

---

### Meta-Review · Area_Chair_dPzJ · 2023-12-06

**Metareview:**

The authors consider the convergence theory of variance-exploding diffusion models, and define a new VE model with modified drift and (unbounded) diffusion term depending on the time t. They prove convergence rates for their VE SDE and VE probability flow ODE under the manifold hypothesis. According to the authors, the main benefit of their parameterization is that the error term from the forward process scales as $e^{-T}$ as opposed to the $1/\text{poly}(T)$ of previous works. They avoid a exp(T) term compared to previous analysis of the PFODE.

Despite an average weak positive rating, reviewers expressed concerns that the VE SDE proposed in this work was only a small modification of that in previous works, there was not a clearly explained benefit over other VE/VP SDE's, and that the numerical experiments were not convincing enough. From my own reading, it seems that the model is essentially a time/space reparameterization that makes the variance explode faster, and that the main claimed benefit of $e^{-T}$ for the forward error follows from this reparameterization; hence the novelty of the analysis (excepting the tangent analysis) is limited, and it is not clear that this improves the overall error.

**Justification For Why Not Higher Score:**

The novelty of the proposed method is limited, and numerical results were not convincing.

**Justification For Why Not Lower Score:**

N/A

---

### Decision · Program_Chairs · 2024-01-16

Reject